# Epigenetic dysregulation of naive CD4+ T-cell activation genes in childhood food allergy

David Martino [1,2], Melanie Neeland[1], Thanh Dang[1], Joanna Cobb[1], Justine Ellis[1], Alice Barnett[1], Mimi Tang[1], Peter Vuillermin[1,3,4], Katrina Allen[1] & Richard Saffery [1]

Food allergy poses a significant clinical and public health burden affecting 2–10% of infants. Using integrated DNA methylation and transcriptomic profiling, we found that polyclonal activation of naive CD4+ T cells through the T cell receptor results in poorer lymphoproliferative responses in children with immunoglobulin E (IgE)-mediated food allergy. Reduced expression of cell cycle-related targets of the E2F and MYC transcription factor networks, and remodeling of DNA methylation at metabolic (*RPTOR*, *PIK3D*, *MAPK1*, *FOXO1*) and inflammatory genes (*IL1R*, *IL18RAP*, *CD82*) underpins this suboptimal response. Infants who fail to resolve food allergy in later childhood exhibit cumulative increases in epigenetic disruption at T cell activation genes and poorer lymphoproliferative responses compared to children who resolved food allergy. Our data indicate epigenetic dysregulation in the early stages of signal transduction through the T cell receptor complex, and likely reflects pathways modified by gene–environment interactions in food allergy.

---

[1] Department of Paediatrics, Murdoch Children's Research Institute, The University of Melbourne, Flemington Road, Parkville, VIC 3052, Australia. [2] Telethon Kids Institute, 100 Roberts Road, Perth, WA, Australia. [3] Barwon Health, Geelong, VIC 3220, Australia. [4] Deakin University, Geelong, VIC 3216, Australia. Correspondence and requests for materials should be addressed to K.A. (email: katrina.allen@mcri.edu.au)

mmunoglobulin E (IgE)-mediated food allergy has become a prominent problem affecting 2–10% of infants[1], and develops from a failure in the immunological pathways mediating oral tolerance during early immune development[2]. The reason that this occurs in some individuals is poorly understood, but related to dysregulation of mucosal immune homeostasis, due to the influence of genetic, epigenetic, and environmental factors on perinatal immune development[3–8]. Immunological studies reveal that many aspects of immune development and function are altered in atopic children and infants with food allergy. Variations in the frequency and function of peripheral blood cell sub-populations are detectable at birth in children who subsequently develop food allergy[9], and are related to pro-inflammatory neo-natal immune responses that persist into early infancy[10]. Atte-nuated T cell signaling[11] and regulatory T cell (Treg)-suppressive function[12] have also been reported among neonates that acquire food allergies in infancy. These initiating mechanisms reflect the early dysregulation of immune development and likely predispose to the eventual failure of mucosal networks that mediate oral tolerance.

Delayed maturation of T cell function is a classical feature of the atopic phenotype[13], and depressed T cell function has been linked with the induction of allergic diseases[14,15], although the mechanisms are unclear. Recent evidence suggests that this extends to food allergy[16,17], and work from our own group[11] and others[9] suggests that suboptimal T cell response capacity to mitogens and allergens is an important pre-morbid factor in the development of food allergy. We have previously described dif-ferences in neonatal total CD4+ T cell activation gene response capacity and proliferative potential in children who eventually develop food allergy in the first year of life[11]. These differences are apparent at birth at an age that is unrelated to allergen exposure and therefore of unknown clinical significance. These data suggest that T cell activation pathways are potentially modified by gene–environment interactions in pregnancy and early life; however, the specific gene and pathways underpinning suboptimal T cell activation have not been defined. It is necessary to define a molecular signature of reduced T cell maturation if future interventions are to be successful. Epigenetics is an attractive area of research as epigenetic modifications can be modified to change gene expression without altering gene sequence. Here we extend this work to focus on naive CD4+ T cells, which are mature multipotent precursors with the capa-city to adopt a range of different T cell effector and memory phenotypes depending on intracellular signaling factors and extracellular cytokine cues. After activation, naive CD4+ T cells establish heritable transcriptional programs that enable progres-sion to short-lived or long-lived effector/memory phenotypes. The initial phase of naive CD4+ T cell priming involves epige-netic remodeling of chromatin that is crucial for mounting effective immune responses, and influencing T cell lineage decisions[18].

In this study, we investigate naive CD4+ T cell activation in a cohort of infants with challenge-confirmed egg allergy, relative to age-matched non-atopic controls. We use genome-wide DNA methylation and transcriptional profiling to delineate molecular pathways of naive T cell responsiveness to activation under neutral (non-differentiating) conditions using bead-bound anti-CD3/anti-CD28 to polyclonally stimulate the canonical T cell receptor signaling pathway. We reasoned that activation under neutral conditions would reveal intrinsic differences in T cell responsiveness independent of external cytokines and antigen presentation cues. By profiling DNA methylation, gene expres-sion, and quantitating cell counts and viability pre-activation and post-activation as a readout of proliferation, we gained insights into naive T cell responsiveness to activation. We compared these

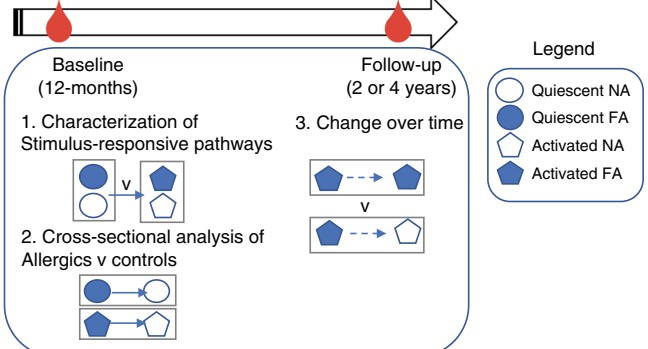

**Fig. 1** Overview of study design and analysis. Clinical assessments occurred at baseline and the allergic subjects were re-assessed several years later at follow-up, at which point 59% had resolved allergy. Bloods were collected at both time points, and frozen for later analysis. Experiments were batched such that naive CD4+ T cells (baseline and follow-up) were flow sorted from bloods prior to cell culture and molecular analyses. NA non-allergic, FA food allergy

data between age-matched cases and controls at 12 months of age (hereafter referred to as baseline) to identify intrinsic differences in gene regulatory pathways associated with food allergy. These analyses were repeated at 2 or 4 years of age (hereafter referred to as follow-up) among the allergic cases, by which time ~60% had outgrown their food allergy. Our data reveal a molecular pattern of T cell activation gene dysregulation in infants with food allergy, associated with poor lymphoproliferative responses.

## Results

**Characterization of T cell stimulus-responsive pathways.** This study was a retrospective analysis of a prospective birth cohort, utilizing both cross-sectional and longitudinal factorial design as outlined in Fig. 1. Characteristics for the subjects selected for this study are shown in Supplementary Table 1. The allergic group consisted of slightly more males (50% of cases vs. 43% of con-trols) and less individuals of parent-reported Caucasian ancestry (88.6% of cases vs. 95% of controls). The allergic group had a higher family history of food allergy (18.1% of cases vs. 0% controls). Children in the control group were more likely to have attended daycare (29.5% of cases vs. 38% of controls), but the groups were otherwise homogenous for commonly studied epi-demiological factors associated with food allergy.

We initially performed an analysis of both the transcriptome and methylome of quiescent vs. naive T cells activated under neutral conditions in all individuals at baseline to characterize pathways associated with T cell activation. In preliminary studies, we determined that 72 h was an optimal time-point for methylation studies as this was the maximum time quiescent naive T cells can rest in culture before substantial culture-induced changes in DNA methylation are detected in the genome-wide scans (Supplementary Figure 1). Activation of T cells for 72 h induced a 2.7-fold expansion compared with un-activated quiescent cells (median quiescent = 500,000, 95% confidence interval (CI) = 385,000–700,000, median activated = $1.36 \times 10^6$, 95% CI = 850,000–$2 \times 10^6$, $P < 0001$), and resulted in the release of large amounts of interleukin-2 (IL-2) and tumor necrosis factor-α (TNF-α), with moderate release of IL-10 and interferon-γ (IFN-γ), with no detectable IL-6 or IL-4 consistent with their profile as T-helper type 0 (Th0) early effector cells[19] (Supple-mentary Figure 2).

Comparing quiescent vs. T cells activated for 72 h under neutral conditions revealed considerable stimulus-responsive changes in gene expression (4154 genes, false discovery rate

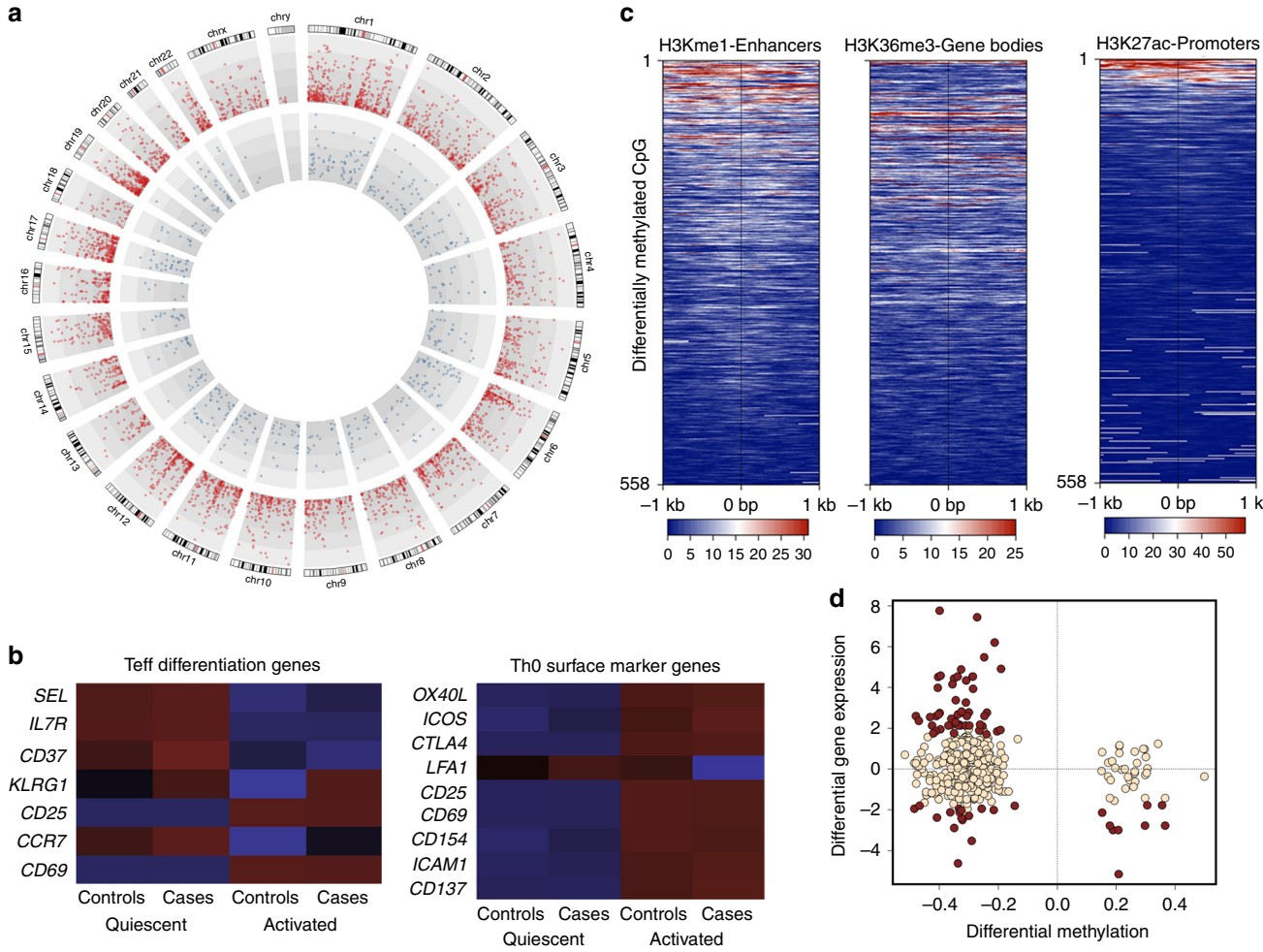

**Fig. 2** Dynamic remodeling of epigenetic landscape in naive T cells following activation. **a** Genome-wide view of the –log 10 P value for 4154 differentially expressed genes (red circles) and 558 differentially methylated CpG (blue circles). **b** Heatmap of average gene expression (rows) for select genes. Cells are colored by the level of expression (blue, low; red, high). **c** Heatmap shows differentially methylated CpGs by genomic location (y-axis) relative to distance from transcriptional start sites (x-axis). Each panel shows enrichment for specific histone marks. **d** Relationship between differential methylation and gene expression for selected genes. X-axis shows the difference in expression (activated minus quiescent cells) as percent methylation ($10^{-2}$). Y-axis shows the log 2 fold change. Points in red were significantly differentially methylated and expressed (remodeled genes) at a genome-wide level. Teff: T effector

(FDR) ≤0.05 and log fold change ≥1.5, Supplementary Data 1) and remodeling of DNA methylation (558 CpG dinucleotides, FDR ≤0.05 and log fold change ≥1.5, Supplementary Data 2) (Fig. 2a). We used gene sets testing of molecular signatures in the hallmark collection[20] to identify biological pathways enriched among the stimulus-responsive genes. Consistent with an activated and highly proliferative state, activated T cells show increased expression of cell cycle-related targets of the E2F transcription factor (TF) network that control cell division and DNA replication, as well as genes regulated by MYC, a proto-oncogene that acts as a molecular "cell division timer" and controls metabolic switching after antigen stimulation[21] (Table 1). Among these pathways were cyclins (*CCNA1, CCNA2, CCNB1, CCNB2, CCND2, CCNE1, CCNE2*) and cyclin-dependent kinases (*CDK1, CDK2, CDK4, CDK6, CDK7*). In accordance with the metabolic requirement to sustain proliferation, we observed upregulation of the MTORC1 pathway and genes involved in glycolysis and fatty acid oxidation. Concomitant with this cell proliferative and metabolic response was expression of JAK-STAT signaling molecules induced by IL-2 in response to activation (*CD25, JAK2, CISH, BCL3, STAT1, STAT3, TSLP, SOCS1–3*) as well as upregulation of various cytokines (*IFNG,*

*IFN1B, IL-2, IL-4, IL6ST, IL13, IL21, IL22*) and cytokine receptors (*IFNLR1, IL11RA, IL12RB2, IL15RA, IL2RA, IL2RB, IL23R, IL3RA, IL6R, IL7R, IL9R*). The latter is indicative of the transition from quiescent to effector cell phenotype. The gene expression profile of activated naive T cells was consistent with an early effector "Th0" phenotype by virtue of their expression of characteristic surface markers (*OX40L, ICOS, CD25, CD69, CD154, ICAM1, CD137*), and T effector gene profile (KLRG1-lo, CD27-lo, CCR7-lo, CD62L-lo, CD25-hi, CD69-hi) (Fig. 2b).

Dynamic changes in the DNA methylation landscape following activation were largely characterized by a widespread loss of DNA methylation at 510/558 (91.3%) regions associated with 220 unique genes, with a modest gain of methylation observed at 48 (8.7%) CpG sites. Using publicly available chromatin immunoprecipitation-sequencing (ChIP-seq) data from primary naive CD4+ T cells, we determined that this widespread loss of methylation was enriched at active enhancer regions marked by the H3K4me1 (enrichment score = 6.4, P = 0.004) and, to a lesser extent, H3K4me3 (enrichment score = 4.1, P = 0.004), H3K27ac (enrichment score = 3.1, P = 0.04), and H3K36me3 (enrichment score = 1.9, P = 0.004) marking actively transcribed promoters and gene bodies, respectively[22] (Fig. 2c, statistics in

**Table 1 Gene sets analysis of differentially expressed and methylated genes in activated vs. quiescent T cells**

**RNA-seq**

| Gene set | NGenes | Direction | *P* value | FDR |
|---|---|---|---|---|
| E2F_TARGETS | 213 | Up | 0.00 | 0.00 |
| MYC_TARGETS_V1 | 215 | Up | 0.00 | 0.00 |
| MTORC1_SIGNALING | 202 | Up | 0.00 | 0.00 |
| G2M_CHECKPOINT | 196 | Up | 0.00 | 0.00 |
| MYC_TARGETS_V2 | 54 | Up | 0.00 | 0.00 |
| OXIDATIVE_PHOSPHORYLATION | 219 | Up | 0.00 | 0.00 |
| FATTY_ACID_METABOLISM | 147 | Up | 0.00 | 0.00 |
| GLYCOLYSIS | 191 | Up | 0.00 | 0.00 |
| UNFOLDED_PROTEIN_RESPONSE | 109 | Up | 0.00 | 0.00 |
| DNA_REPAIR | 158 | Up | 0.00 | 0.00 |
| CHOLESTEROL_HOMEOSTASIS | 75 | Up | 0.00 | 0.00 |
| ADIPOGENESIS | 195 | Up | 0.00 | 0.00 |
| IL2_STAT5_SIGNALING | 205 | Up | 0.00 | 0.00 |
| TNFA_SIGNALING_VIA_NFKB | 201 | Up | 0.00 | 0.00 |
| HYPOXIA | 189 | Up | 0.00 | 0.00 |
| APOPTOSIS | 165 | Up | 0.00 | 0.00 |
| REACTIVE_OXIGEN_SPECIES_PATHWAY | 50 | Up | 0.00 | 0.00 |
| UV_RESPONSE_UP | 169 | Up | 0.00 | 0.00 |
| INTERFERON_GAMMA_RESPONSE | 245 | Up | 0.00 | 0.00 |
| PEROXISOME | 92 | Up | 0.00 | 0.00 |
| **DNA methylation** | | | | |
| **Gene sets** | **NGenes** | **DM** | ***P* value** | **FDR** |
| APOPTOSIS | 161 | 6 | 0.01 | 0.06 |
| MITOTIC_SPINDLE | 200 | 7 | 0.01 | 0.05 |
| COMPLEMENT | 200 | 7 | 0.01 | 0.05 |
| HEME_METABOLISM | 200 | 7 | 0.01 | 0.05 |
| PI3K_AKT_MTOR_SIGNALING | 105 | 5 | 0.01 | 0.05 |
| TGF_BETA_SIGNALING | 54 | 4 | 0.00 | 0.03 |
| IL2_STAT5_SIGNALING | 200 | 8 | 0.00 | 0.03 |
| INFLAMMATORY_RESPONSE | 200 | 9 | 0.00 | 0.01 |
| TNFA_SIGNALING_VIA_NFKB | 200 | 10 | 0.00 | 0.00 |
| ALLOGRAFT_REJECTION | 200 | 10 | 0.00 | 0.00 |
| UV_RESPONSE_DN | 144 | 10 | 0.00 | 0.00 |

DM: differentially methylated; FDR: false discovery rate, *P* value

Supplementary Table 2). Integrated analysis of expression and methylation data sets showed a clear relationship between differentially methylated CpG sites and changes in gene expression (Fig. 2d). In many cases, loss of methylation correlated with upregulation of the expression of the corresponding gene and vice versa. Generally, changes in gene expression were modest, although a core set of 59 unique genes were both differentially methylated and expressed at the genome-wide level of significance, and we refer to these as "differentially remodeled" (Fig. 2d, points in red). This is consistent with the establishment of a permissive epigenetic state associated with a shift in stable gene expression profile. Gene set analysis of this subset of "remodeled" genes indicated enriched targets of the nuclear factor of activated T cell TF network (FDR *P* value = $2.76 \times 10^{-76}$) involved in the IL-2-STAT5 signaling pathway (FDR *P* value = $1.34 \times 10^{-3}$, *IL2RA*, *RORA*, *BCL21*, *GALM*), TNFA-NFKB signaling pathway (FDR *P* value = $1.34 \times 10^{-3}$, *ATF3*, *NFKB1*, *NR4A3*, *B4GALT5*), and the IFN-γ response (FDR *P* value = $1.34 \times 10^{-3}$, *NFKB1*, *CASP7*, *CD274*).

**Case–control analysis of allergic infants**. We next explored whether T cell activation-induced changes in the epigenetic and transcriptional landscape were related to food allergy status in samples collected at baseline (12-month infants). Unsupervised principal component (PC) clustering analysis of the 4154 differentially expressed genes, and the 558 differentially methylated loci revealed clustering according to both T cell activation status (PC1) and food allergy status (PC2) (Fig. 3a, b). Clustering by food allergy status was substantially more evident among activated cells. This suggested that T cell activation gene networks were at least partially related to food allergy status.

Functional measures of T cell activation also suggested that food allergic children exhibited an attenuated proliferative response capacity compared with age-matched non-atopic counterparts (median non-allergics = $3 \times 10^6$, median allergics = $7 \times 10^5$, *P* < 0.0001, Fig. 4a). This hypo-proliferative response was not explained by a higher rate of cell death in the allergic group (Fig. 4a), nor was it explained by differences in the expression of CD3 at the apical surface prior to activation (Supplementary Figure 3). Analysis of secreted cytokines indicated the IL-2 response, which is coupled to T cell receptor signaling, was intact among allergic individuals arguing against T cell anergy as a potential explanation. We also noted that there was evidence for higher number of IL-10 (*P* exact = 0.0018) and IFN-γ (*P* exact = 0.024) responders in the activated cells from allergics (Fig. 4b).

To further quantify molecular changes associated with food allergy, we compared genome-wide DNA methylation and gene expression data sets between allergic cases and non-atopic controls. In activated T cells, we detected substantial differential expression at 1412 genes (FDR ≤0.05, Supplementary Data 3) and differential methylation 189 CpG sites (FDR ≤0.05, Supplementary Data 4) (Fig. 4c). There was weak evidence that these

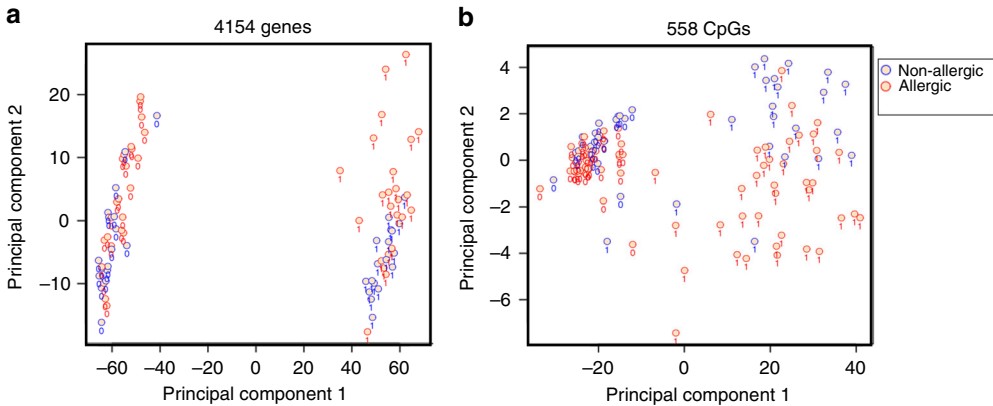

**Fig. 3** Unsupervised clustering analysis of T cell activation genes reveals variation by food allergy status. Principal component clustering analysis of 12-month patient samples based on 4154 T cell activation response genes (**a**) and 558 CpGs (**b**). Samples are labeled according to activation status, 0 = quiescent, 1 = activated, and colored according to phenotype, blue = non-allergic, red = food allergy

**Fig. 4** T cell hypo-responsiveness in allergic children is underpinned by altered remodeling of metabolic and inflammatory genes. **a** Proliferative responses and cell viability following T cell activation. Data are expressed as fold change calculated as post–pre-activation cell counts, with bars showing median and interquartile range. Groups were compared using the Mann–Whitney test. ***$P < 0.001$. NA.con non-allergic quiescent, NA.act non-allergic activated, FA. con food allergy quiescent, FA.act food allergy activated. **b** Supernatant cytokines, with data expressed as median with interquartile range. Groups were compared using the Mann–Whitney test. **c** Genome-wide view of the –log 10 $P$ value for 1412 differentially expressed genes (red circles) and 189 differentially methylated CpG (blue circles). **d** Similarity (Forbes coefficient) between allergy-dmrs and regions marked by activating histone modifications in naive and effector/memory cells, respectively. Similarity calculated by ratio of observed/expected overlap between these regulatory regions and allergy-dmrs. **e** Relationship between differential methylation and gene expression. X-axis shows delta value expressed as percent methylation ($10^{-2}$) for the comparison of cases–controls. Y-axis shows the log 2 fold change. Points in red were differentially methylated and expressed (remodeled genes) at the genome-wide level. Eff effector, mem memory

**Table 2 Gene sets analysis of differentially expressed and methylated genes in allergics vs. controls at baseline**

**RNA-seq**

| Gene set | NGenes | Direction | P value | FDR |
|---|---|---|---|---|
| E2F_TARGETS | 213 | Down | 0.00 | 0.00 |
| G2M_CHECKPOINT | 196 | Down | 0.00 | 0.00 |
| MYC_TARGETS_V1 | 215 | Down | 0.00 | 0.03 |
| SPERMATOGENESIS | 104 | Down | 0.09 | 0.22 |
| MYOGENESIS | 151 | Up | 0.00 | 0.01 |
| APICAL_JUNCTION | 185 | Up | 0.00 | 0.03 |
| CHOLESTEROL_HOMEOSTASIS | 75 | Up | 0.00 | 0.03 |
| TNFA_SIGNALING_VIA_NFKB | 201 | Up | 0.01 | 0.03 |
| WNT_BETA_CATENIN_SIGNALING | 40 | Up | 0.01 | 0.03 |
| IL2_STAT5_SIGNALING | 205 | Up | 0.01 | 0.03 |
| EPITHELIAL_MESENCHYMAL_TRANSITION | 152 | Up | 0.01 | 0.03 |
| UV_RESPONSE_UP | 169 | Up | 0.01 | 0.03 |

**DNA methylation**

| Gene set | NGenes | DM | P value | FDR |
|---|---|---|---|---|
| UV_RESPONSE_DN | 144 | 10 | 0.00 | 0.00 |
| TNFA_SIGNALING_VIA_NFKB | 200 | 10 | 0.00 | 0.00 |
| ALLOGRAFT_REJECTION | 200 | 10 | 0.00 | 0.00 |
| INFLAMMATORY_RESPONSE | 200 | 9 | 0.00 | 0.01 |
| IL2_STAT5_SIGNALING | 200 | 8 | 0.00 | 0.03 |
| TGF_BETA_SIGNALING | 54 | 4 | 0.00 | 0.03 |
| PI3K_AKT_MTOR_SIGNALING | 105 | 5 | 0.01 | 0.05 |

DM: differentially methylated; FDR: false discovery rate, P value

activation-induced genomic changes were explained by pre-existing differences in gene expression (two genes, FDR ≤0.05) or DNA methylation (2 CpG, FDR ≤0.05) when comparing quiescent cells between cases and controls (Supplementary Table 3). This indicated that, in general, the basal state of resting naive T cells was similar between the groups, but they largely differed in their capacity to invoke the T cell activation program. In support of this, we found that these signatures of food allergy partially overlap T cell activation pathways, as 228/1412 (16%) of the food allergy genes, and 33/156 (21%) of the food allergy CpGs were also identified in the previous analysis of T cell activation.

In the activated T cells, there were far more differentially expressed genes compared with differentially methylated CpG sites (Fig. 4c). A striking feature of these data was the over-representation of differentially expressed genes on chromosome 19 (Fig. 4c) that were highly enriched for C2H2-like family of zinc finger proteins (Interpro: IPR015880, adjusted $P = 1.6 \times 10^{-57}$), which were down-regulated in activated T cells in the allergic group. This family includes DNA-binding TFs with known functions in T cell development (*BCL11B*[23]) and T cell exhaustion[24]. Gene sets testing revealed that targets of the E2F and MYC TF networks and genes involved in the G2/M cell cycle checkpoint transition were down-regulated among children with food allergy (Table 2). Concomitantly, pathways related to structural components of the cell cytoskeleton involved in the contractile response (myogenesis, apical junction, epithelial transition), signal transduction (STAT and WNT signaling), and fatty acid metabolism (cholesterol homeostasis) were significantly upregulated. Taken together, these data delineate a relationship between reduced expression of cell cycle genes, signal transduction, and metabolic gene response programs.

Among the 189 differentially methylated sites identified in case–control comparisons of activated T cells, we observed that food allergy was associated with substantial loss of methylation at 89% of these sites (169/189), and gain of methylation at 11% (20/189) CpG sites. Loss of methylation was localized to 112 unique genes, with multiple hits observed in some genes (*CDK6*, *GPR55*, *RAD51L1*, *RPTOR*, *CD96*, *XXYLT1*; Supplementary Data 4). Gain of methylation was localized to 14 unique genes (Supplementary

Data 4). Differential methylation was enriched within the gene body rather than promoters or CpG islands (Supplementary Figure 4). We downloaded publicly available DNAse hypersensitivity and ChIP-seq data from the ENCODE consortium from CD4+ naive and effector/memory T cell populations to determine whether allergy-associated differentially methylated regions (allergy-dmrs) were enriched at lineage-defining active chromatin regions. We also included data from monocytes as a negative control. We found that allergy-dmrs were enriched within regions of accessible chromatin in effector T cells and depleted in regions of accessible chromatin in naive T cells and monocytes (Supplementary Figure 5). ChIP-seq data supported this as allergy-dmrs were enriched for activating marks (H3K4me1, H3K27ac) in effector/memory cells relative to naive cells (Fig. 4d). Integrative analysis of methylation and gene expression data sets revealed that differentially methylated CpG sites were generally anti-correlated with changes in gene expression (Fig. 4e), and by comparing overlaps between differentially methylated and expressed data sets, we identified a core set of 24 "differentially remodeled" genes, indicative of those that had undergone a state change toward a more stable expression profile (Table 3). Of particular interest among these were genes involved in the metabolic regulation of oxidative phosphorylation and glycolysis (*RPTOR*, *PIK3D*, *MAPK1*, *FOXO1*) and inflammatory genes (*IL1R*, *IL18RAP*, *CD82*), which were upregulated in the allergics (Fig. 4e). In contrast, the Lck-interacting transmembrane adapter 1 (*LIME1*) transcript, which encodes an adaptor protein that links T cell receptor stimulation to its downstream signaling pathway, was hyper-methylated and suppressed in the allergics (Fig. 4e).

**Polymorphisms at specific loci do not influence methylation.** Given previous published evidence that food allergy-associated dmrs can mediate the effect of genetic variation[5], we sought to determine whether our differentially remodeled T cell activation genes might be driven by local polymorphism. To do this, we analyzed single-nucleotide polymorphism (SNP)-array data available for 42 individuals (14 cases and 28 controls) in this

**Table 3 Differentially remodeled genes associated with food allergy at 12 months**

| | DNA methylation | | | | | | Gene expression | | | | | |
|---|---|---|---|---|---|---|---|---|---|---|---|---|
| | ProbeID | Chr | Start | Stop | Deltabeta | FDR | Chr | Start | Stop | Log FC | FDR | Gene |
| 1 | cg12592365 | Chr17 | 78,765,948 | 78,765,949 | −0.05 | 0.00 | Chr17 | 78,518,619 | 78,940,171 | 0.439240548 | 0.00 | *RPTOR* |
| 2 | cg02660643 | Chr16 | 87,491,587 | 87,491,588 | −0.05 | 0.00 | Chr16 | 87,439,852 | 87,525,651 | 0.411181941 | 0.01 | *ZCCHC14* |
| 3 | cg27403618 | Chr1 | 209,907,350 | 209,907,351 | −0.04 | 0.00 | Chr1 | 209,859,510 | 209,908,295 | 1.342450486 | 0.04 | *HSD11B1* |
| 4 | cg22152180 | Chr1 | 172,558,180 | 172,558,181 | −0.04 | 0.00 | Chr1 | 172,501,489 | 172,580,971 | −0.221601774 | 0.02 | *SUCO* |
| 5 | cg26319797 | Chr2 | 238,904,315 | 238,904,316 | −0.03 | 0.00 | Chr2 | 238,875,469 | 238,951,236 | 0.424532964 | 0.00 | *UBE2F* |
| 6 | cg25298754 | Chr3 | 111,314,102 | 111,314,103 | −0.03 | 0.00 | Chr3 | 111,311,747 | 111,314,290 | 0.919262265 | 0.03 | *ZBED2* |
| 7 | cg09775648 | Chr2 | 190,309,186 | 190,309,187 | −0.05 | 0.01 | Chr2 | 190,306,159 | 190,340,291 | 0.384026907 | 0.03 | *WDR75* |
| 8 | cg22992505 | Chr2 | 158,712,427 | 158,712,428 | −0.04 | 0.01 | Chr2 | 158,592,958 | 158732,374 | 0.295193799 | 0.05 | *ACVR1* |
| 9 | cg00545580 | Chr17 | 78,571,214 | 78,571,215 | −0.05 | 0.01 | Chr17 | 78,518,619 | 78,940,171 | 0.439240548 | 0.00 | *RPTOR* |
| 10 | cg05175803 | Chr11 | 44,597,119 | 44,597,120 | −0.01 | 0.01 | Chr11 | 44,585,977 | 44,641,913 | 0.445341065 | 0.01 | *CD82* |
| 11 | cg24278087 | Chr11 | 65,904,369 | 65,904,370 | −0.03 | 0.01 | Chr11 | 65,837,834 | 66,012,218 | 0.409525829 | 0.00 | *PACS1* |
| 12 | cg16001422 | Chr8 | 145,022,842 | 145,022,843 | −0.04 | 0.02 | Chr8 | 144,989,321 | 14,5050,902 | 0.554875897 | 0.02 | *PLEC* |
| 13 | cg03116016 | Chr16 | 15,583,606 | 15,583,607 | −0.03 | 0.02 | Chr16 | 15,528,152 | 15,718,885 | 0.920652897 | 0.01 | *C16orf45* |
| 14 | cg25279613 | Chr7 | 24,956,523 | 24,956,524 | −0.06 | 0.02 | Chr7 | 24,836,158 | 25,021,253 | 0.276014688 | 0.03 | *OSBPL3* |
| 15 | cg00957665 | Chr10 | 104,406,345 | 104,406,346 | −0.01 | 0.02 | Chr10 | 104,404,253 | 104,418,164 | 0.496130053 | 0.00 | *TRIM8* |
| 16 | cg25867265 | Chr1 | 66,737,176 | 66,737,177 | 0.06 | 0.03 | Chr1 | 66,258,197 | 66,840,259 | −0.443748368 | 0.05 | *PDE4B* |
| 17 | cg20995327 | Chr14 | 99,684,844 | 99,684,845 | −0.03 | 0.03 | Chr14 | 99,635,624 | 99,737,861 | −0.250717631 | 0.04 | *BCL11B* |
| 18 | cg17355385 | Chr20 | 62,368,837 | 62,368,838 | 0.05 | 0.03 | Chr20 | 62,366,815 | 62,370,456 | −0.634130427 | 0.03 | *LIME1* |
| 19 | cg20930706 | Chr12 | 12,595,583 | 12,595,584 | −0.02 | 0.03 | Chr12 | 12,510,013 | 12,619,840 | 0.347785134 | 0.01 | *LOH12CR1* |
| 20 | cg14174367 | Chr1 | 202,128,508 | 202,128,509 | −0.03 | 0.04 | Chr1 | 202,116,141 | 202,130,716 | 0.239397286 | 0.05 | *PTPN7* |
| 21 | cg24323726 | Chr3 | 111,314,186 | 111,314,187 | −0.03 | 0.04 | Chr3 | 111,311,747 | 111,314,290 | 0.919262265 | 0.03 | *ZBED2* |
| 22 | cg21410897 | Chr2 | 69,735,185 | 69,735,186 | −0.02 | 0.04 | Chr2 | 69,688,532 | 69,901,481 | 0.284488561 | 0.03 | *AAK1* |
| 23 | cg11903019 | Chr6 | 154,560,711 | 154,560,712 | −0.01 | 0.04 | Chr6 | 154,475,631 | 154,677,926 | −0.471902349 | 0.01 | *IPCEF1* |
| 24 | cg04599190 | Chr5 | 66,306,763 | 66,306,764 | −0.02 | 0.04 | Chr5 | 65,892,176 | 66,465,423 | 0.382960527 | 0.01 | *MAST4* |
| 25 | cg10502206 | Chr2 | 145,182,344 | 145,182,345 | −0.02 | 0.04 | Chr2 | 145,141,648 | 145,282,147 | 0.778825531 | 0.05 | *ZEB2* |
| 26 | cg15928524 | Chr17 | 17,599,543 | 17,599,544 | −0.04 | 0.05 | Chr17 | 17,584,787 | 17,714,767 | 0.312294786 | 0.02 | *RAI1* |

FDR: false discovery rate *P* value

study and performed association testing of individual genotypes and food allergy, and also computed linear regression models for each SNP/gene and SNP/CpG pair. We limited our analysis to all SNPs genotyped on the array within a heuristic 10-kb window up and downstream of the core set of 24 differentially remodeled genes[25], as well as within genes encoding the de novo DNA methyltransferase enzymes DNMT3A and DNMT3B and previously published food allergy SNPs annotated in the genome-wide association study and SNPedia catalogs[5,26]. We also queried the largest blood methylation quantitative trait loci (mQTL) database[27] for any SNPs that could potentially influence methylation at the 24 genes of interest. In total, 87 high-quality SNPs were tested for association with food allergy after adjusting for ancestry using a heuristic un-adjusted $P < 0.1$ threshold. Of these, 3/87 SNPs in the *RPTOR* gene (rs9906827, rs2672886, rs9908768) showed weak evidence of an association ($P = 0.070$, $P = 0.047$, $P = 0.093$) with food allergy (Supplementary Table 4). This suggested that patterns of differential methylation/gene expression at *RPTOR* may be influenced by genetic risk variants. To determine this, we tested for associations between SNP/gene pairs and SNP/CpG pairs by extracting *RPTOR* transcripts and CpG methylation levels from the corresponding data sets. We found no evidence for an association between the three SNPs and *RPTOR* transcript levels (rs9906827, $P = 0.85$; rs2672886, $P = 0.56$; rs9908768, $P = 0.48$). In total, 504 probes on the DNA methylation microarray were annotated to *RPTOR* and we computed regression models between SNP/CpG pairs generating 1512 $P$ value associations. We found evidence for six associations (FDR $P$ value <0.05) indicating that methylation patterns at these six loci were under the influence of genetic variation. When we restricted this analysis to just the food allergy-associated dmps in *RPTOR* (cg12592365, cg00545580; Supplementary Data 4), we found no evidence for an association (Supplementary Figure 6), suggesting that

while local DNA methylation profiles at *RPTOR* may be influenced by genotype, we did not find evidence that loss of methylation at *RPTOR* associated with food allergy was substantially influenced by genetic variation within the SNPs tested in this cohort.

**Persistence of food allergy in childhood.** Within this cohort, 26 of the egg allergic individuals (59% of cases) naturally acquired tolerance to egg by the time they were assessed at follow-up. This sample size was underpowered to identify genomic changes associated with the development of clinical tolerance at the genome-wide level, so this was not carried out. Rather, we performed a longitudinal analysis of the loci associated with food allergy at baseline to determine whether these changes were differentially modified from baseline to follow-up. We performed linear regression in the DNA methylation and gene expression data sets separately, with phenotype at follow-up (persistent or resolved) as the outcome variable, and DNA methylation/gene expression as the predictor. We also assessed proliferative responses and cytokine production.

We found that the majority of (24/26) loci associated with the core set of "differentially remodeled" genes exhibited a statistically significant (FDR ≤0.05, Table 4) change in DNA methylation from baseline to follow-up among children with persistent food allergy, while epigenetic changes in the transient group were stable (Fig. 5a). This suggested cumulative epigenetic disruption and was associated with a trend for poorer lymphoproliferative responses compared to children who resolved egg allergy (mean resolved: 906,482, 95% CI: 466,333–1.34 × 10⁶ vs. mean allergic: 395,375, 95% CI: −280,088 to 107 × 10⁶) at follow-up (Fig. 5b). Production of IFNγ was higher after activation among children with persistent allergy (Fig. 5c).

**Table 4 Longitudinal analysis of differentially remodeled genes**

| ProbeID | Chr | Pos | Persistent allergics (n = 49) | | | | Transient allergics (n = 115) | | | | Gene |
|---------|-----|-----|-------------------|-------------------|---------------|---------------------|-------------------|-------------------|---------------|---------------------|------|
| | | | Deltabeta (FU-BL) | Adj.P value deltabeta | Log FC (FU-BL) | Adj. P value log FC | Deltabeta (FU-BL) | Adj. P value deltabeta | Log FC (FU-BL) | Adj. P. val log FC | |
| cg20995327 | Chr12 | 12,595,583 | 0.172 | 0.000 | −0.248 | 0.178 | 0.025 | 0.051 | 0.145 | 0.296 | LOH12CR1 |
| cg22152180 | Chr2 | 238,904,315 | 0.079 | 0.000 | 0.134 | 0.536 | 0.026 | 0.095 | 0.125 | 0.296 | UBE2F |
| cg09775648 | Chr3 | 111,314,102 | 0.115 | 0.000 | −0.179 | 0.611 | 0.058 | 0.095 | −0.425 | 0.001 | ZBED2 |
| cg26319797 | Chr2 | 190,309,186 | 0.148 | 0.000 | −0.529 | 0.012 | 0.030 | 0.095 | −0.114 | 0.467 | WDR75 |
| cg24278087 | Chr11 | 44,597,119 | 0.072 | 0.000 | −0.196 | 0.428 | 0.014 | 0.101 | −0.088 | 0.591 | CD82 |
| cg05175803 | Chr17 | 78,765,948 | 0.136 | 0.000 | −0.230 | 0.323 | 0.025 | 0.141 | −0.154 | 0.390 | RPTOR |
| cg27403618 | Chr1 | 209,907,350 | 0.078 | 0.000 | −0.549 | 0.428 | 0.027 | 0.160 | −0.213 | 0.627 | HSD11B1 |
| cg12592365 | Chr16 | 15,583,606 | 0.100 | 0.000 | −0.408 | 0.345 | 0.042 | 0.160 | −0.075 | 0.612 | C16orf45 |
| cg02660643 | Chr2 | 69,735,185 | 0.097 | 0.000 | 0.023 | 0.931 | 0.016 | 0.160 | −0.033 | 0.758 | AAK1 |
| cg14174367 | Chr17 | 17,599,543 | 0.085 | 0.001 | −0.189 | 0.428 | 0.027 | 0.197 | −0.289 | 0.001 | RAI1 |
| cg25279613 | Chr8 | 145,022,842 | 0.056 | 0.001 | 0.245 | 0.498 | 0.029 | 0.197 | −0.104 | 0.467 | PLEC |
| cg17355385 | Chr2 | 158,712,427 | 0.068 | 0.001 | 0.030 | 0.931 | −0.022 | 0.245 | −0.089 | 0.758 | ACVR1 |
| cg20930706 | Chr1 | 172,558,180 | 0.055 | 0.001 | 0.141 | 0.428 | 0.026 | 0.455 | −0.071 | 0.607 | SUCO |
| cg15928524 | Chr3 | 111,314,186 | 0.055 | 0.001 | −0.179 | 0.611 | 0.016 | 0.464 | −0.162 | 0.296 | ZBED2 |
| cg00545580 | Chr6 | 154,560,711 | 0.092 | 0.002 | 0.695 | 0.012 | 0.014 | 0.556 | −0.075 | 0.612 | IPCEF1 |
| cg22992505 | Chr20 | 62,368,837 | −0.073 | 0.003 | −0.059 | 0.931 | 0.011 | 0.568 | −0.034 | 0.758 | LIME1 |
| cg03116016 | Chr7 | 24,956,523 | 0.079 | 0.004 | 0.000 | 0.998 | 0.007 | 0.805 | 0.046 | 0.774 | OSBPL3 |
| cg24323726 | Chr11 | 65,904,369 | 0.024 | 0.009 | −0.181 | 0.428 | −0.006 | 0.805 | 0.089 | 0.758 | PACS1 |
| cg04599190 | Chr17 | 78,571,214 | 0.061 | 0.018 | −0.230 | 0.323 | 0.002 | 0.805 | −0.039 | 0.758 | RPTOR |
| cg00957665 | Chr14 | 99,684,844 | 0.029 | 0.019 | 0.049 | 0.850 | 0.005 | 0.805 | −0.228 | 0.027 | BCL11B |
| cg25867265 | Chr2 | 145,182,344 | 0.034 | 0.023 | −0.228 | 0.536 | −0.005 | 0.805 | 0.295 | 0.106 | ZEB2 |
| cg11903019 | Chr1 | 202,128,508 | 0.051 | 0.025 | −0.428 | 0.006 | −0.009 | 0.805 | 0.349 | 0.053 | PTPN7 |
| cg16001422 | Chr16 | 87,491,587 | 0.026 | 0.029 | 0.255 | 0.323 | 0.004 | 0.876 | −0.102 | 0.719 | ZCCHC14 |
| cg21410897 | Chr10 | 104,406,345 | 0.033 | 0.040 | −0.323 | 0.049 | 0.003 | 0.876 | 0.124 | 0.390 | TRIM8 |
| cg10502206 | Chr5 | 66,306,763 | 0.007 | 0.384 | −0.168 | 0.464 | 0.002 | 0.896 | −0.081 | 0.758 | MAST4 |
| cg25298754 | Chr1 | 66,737,176 | 0.015 | 0.469 | 0.624 | 0.018 | −0.001 | 0.999 | 0.089 | 0.758 | PDE4B |

Change in DNA methylation (deltabeta) and gene expression (log FC) from baseline to follow-up are shown
BL: baseline; FU: follow-up

## Discussion

Allergic diseases are thought to arise due to complex gene–environment interactions impacting upon normal immune development[28], and reduced or altered functional capacity of T cell responses is a consistently observed manifestation of the atopic phenotype[13,29,30]. Our results indicate that naive T cells from children with food allergy, although they appear immuno-competent in relation to IL-2 production, exhibit an intrinsic molecular defect during the early state of priming, and depressed capacity for proliferation. Previous studies have speculated that this may be attributed to intrinsic defects in T cells themselves, or reduced/altered functional capacity of accessory cells such as dendritic cells that prime T cell responses[31]. In the current study, we isolated a homogenous population of antigen-naive CD4+ T cells and activated them independently of accessory cells under non-polarizing conditions. Our data are consistent with an intrinsic defect in naive T cells that is widespread and polyclonal in nature, and likely reflects the different kinetics or trajectory of normal immune development in allergic disease.

Activated T cells reprogram specific metabolic and mitotic gene networks that are subsequently refined by epigenetic modifications[32]. Our analysis of the epigenetically regulated transcriptional response points to gene networks activated via E2F and MYC, and mTOR metabolic programs as markedly altered in children with food allergy and delineates these as key pathways modified by gene–environment interactions. These TFs integrate activating and inhibitory signals to mediate thresholds of activation[21,33], transforming quiescent T cells into highly mitotic proliferating cells. In addition, the striking over-representation of zinc finger genes of the highly conserved C2H2 class on

chromosome 19 emerged as the strongest enrichment signal among the list of down-regulated genes in the allergics. This family of TFs preferentially recognize methylated DNA acting mostly as chromatin-modulating transcriptional repressors of methylated DNA templates[34]. The significance of this finding is unclear. One previous study also reported differential expression of the C2H2 transcription factors between memory and exhausted CD8+ T cells[24]. We did not see evidence of upregulation of classical inhibitory markers associated with T cell exhaustion. This finding may therefore reflect a lack of coordinated transcriptional regulation of quiescence genes.

These findings extend our previous work reporting on attenuated lymphoproliferative responses in bulk CD4+ cells from the cord blood of children with IgE-mediated food allergy[11], and confirms that maturational defects in TCR signaling extend to naive CD4+ T cell precursors. These and other recent data from cord blood studies of immune function in children who develop IgE food allergy[9,12,35] imply that faulty immune development precedes the clinical manifestation of disease and is clearly important in disease etiology. How the molecular changes described here are related to the maturation and development of postnatal immune responses to allergens, or the development of functionally unique pathogenic subtypes of allergen-reactive T cells[16] is unclear, and needs to be the focus on ongoing investigation. A key finding from our study was that differential methylation associated with food allergy was enriched at lineage-defining regions of open chromatin in effector and memory T cells (Figure S5) and overlapped activating histone marks in terminally differentiated T cell phenotypes (Fig. 4d). This implies that changes in DNA methylation levels seen in allergic children

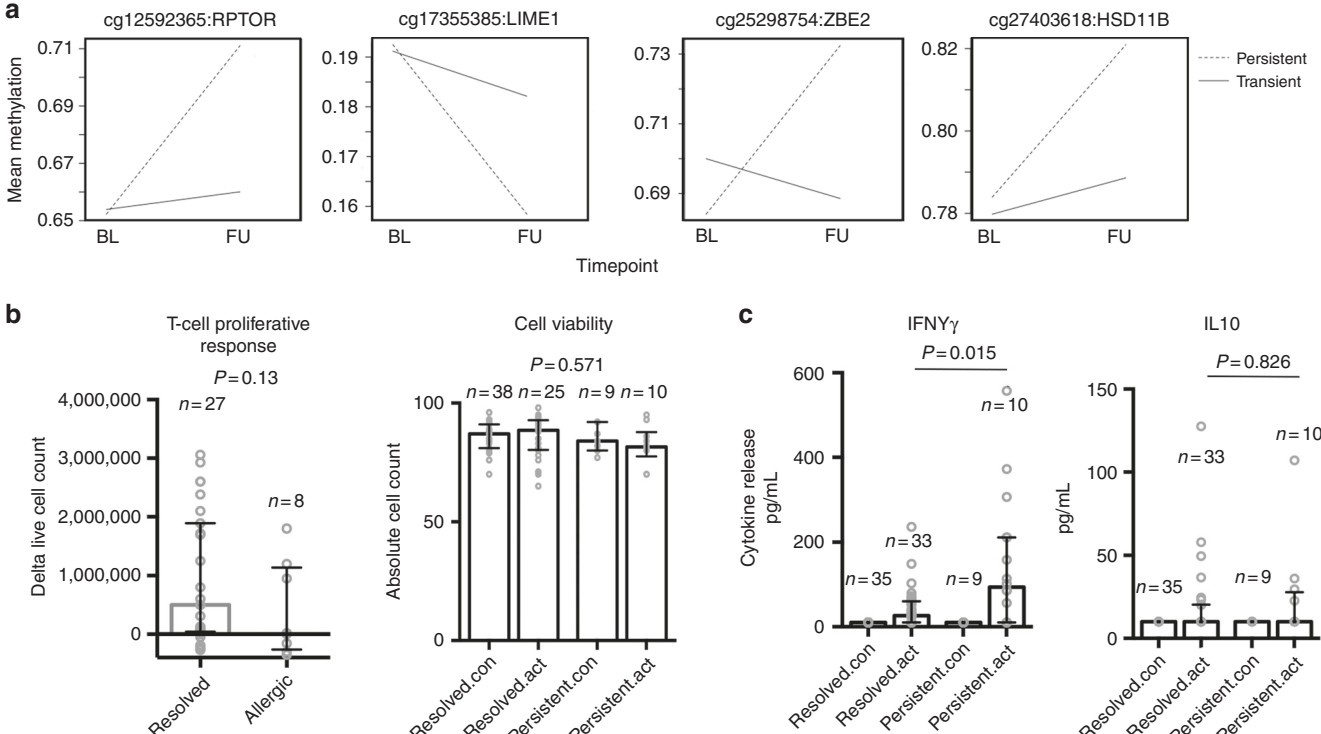

**Fig. 5** Analysis of allergy candidates at follow-up between persistent and resolved allergics. **a** The top panels show representative interactions plots for 4 of the 24 remodeled candidates identified at baseline. Children with persistent allergy show cumulative epigenetic perturbation with age. Filled and dashed lines represent the trajectory for the change over time in mean methylation levels ($10^{-2}$). Probe Ids and gene names are shown above, and full statistics are provided in Table 4. **b** Proliferative responses and cell viability following T cell activation at follow-up are shown as median with interquartile range. **c** Cytokine production was expressed as median with interquartile range. For **b** and **c** groups were compared using the Mann–Whitney test. Exact $P$ values are shown. For cell viability, exact $P$ value was derived from the one-way ANOVA test

during naive T cell activation may alter T cell fate decisions, which may skew T cell phenotypes at a very early point, but this requires more detailed investigation.

Our study adds to the growing body of literature describing epigenetic disruption in cow's milk allergy[4,36–38] and peanut allergy[39,40]. The question of whether our data reflect the genetic atopic predisposition is an important one since historical studies clearly show that atopy affects postnatal maturation of T cell competence[13]. We note that 18% of our cases had a reported family history of food allergy, and thus we made use of available SNP-array genotyping data to perform an association analysis on a limited number of "modified" food allergy genes, but did not find strong evidence to suggest that genetic variation in these genes was driving changes in DNA methylation. Moreover, the longitudinal analysis suggests the accumulation of epigenetic change throughout childhood uniquely among the group of children who failed to resolve food allergy, which would be inconsistent with genetic effects on methylation that are reported to be fairly stable over time[27]. This should be interpreted cautiously and in the context of limited study power, as extensive investigation of the effect of genetic atopic predisposition on the epigenetic regulation of gene expression was beyond the scope of this study. Large comprehensive studies are needed to fully address this. We did not detect differences in the patterning of DNA methylation or gene expression in quiescent cells that might explain the altered T cell responsiveness seen in the food allergic infants. This might suggest that other epigenetic or post-translational modifications are relevant.

Our study has several strengths. We started with a homogenous population of naive CD4+ T cells from a very well-characterized cohort of infants with diagnostic food challenge outcomes. We examined T cell function using a highly standardized protocol across multiple resolutions of the genome, transcriptome, and epigenome. Combining these data enabled new insights into the transcriptional and epigenetic basis of T cell hypo-responsiveness. The longitudinal design allowed us to determine that the pathways described here were continually modified in children whom failed to resolve food allergy.

Limitations included relatively small sample size. We were also unable to include a cell division tracking dye into the flow cytometry antibody panels, and therefore unable to accurately assess the number of cell divisions occurring in the 72 h following activation or run detailed time-course experiments. In addition, cell death was assessed by Trypan blue staining rather than markers of apoptosis such as Annexin-V. Our study is not able to address the question of whether T cell activation is suboptimal in antigen-specific naive T cell populations, although we speculate that this is the case. Future studies should seek to address the epigenetic status of allergen-specific clones using peptide epitopes loaded onto major histocompatibility complex class II tetramers.

In summary, this study revealed that food allergy in infancy is associated with naive T cell hypo-responsiveness to activation, which is related to epigenetic changes in metabolic and immunological genes. Studies are now required to determine whether the pathways identified here influence T cell fate decisions and may skew the naive T cell response at an early time-point, predisposing to the development of pathogenic CD4+ T cell subsets.

## Methods

**Study population and clinical data collection.** Subjects were participants in the population-based HealthNuts cohort study[41–43] recruited through community immunization clinics, and clinically assessed at the Royal Children's Hospital in

Melbourne, Australia, with informed consent and approved by the Royal Children's Human Research Ethics Committee (ref. no. 27047). All subjects were aged between 11 and 15 months at the time of initial assessment (baseline), and underwent skin prick testing to egg white, peanut, sesame, and one of two other foods (cow's milk or shrimp) and oral food challenges (OFCs) using predetermined challenge criteria[42]. Allergic subjects ($n = 44$) were selected for this sub-study if they were mono-sensitized and clinically reactive to egg at 12 months of age (baseline), while age-matched controls ($n = 21$) were non-atopic to any of the allergens tested. The allergic group subsequently returned for follow-up clinical assessments at 2 or 4 years of age using identical criteria, at which time 26/44 (59%) children had outgrown their egg allergy, clinically defined as "transient egg allergy" while 18/44 (41%) remained allergic (denoted as "persistent egg allergy"). Blood was collected into a sodium heparin tube (Sarstedt, Inc, Newton, NC, USA) 1–2 h after the OFC and peripheral blood mononuclear cells (PBMCs) purified using Ficoll-Paque density gradient centrifugation within 2 h of blood collection. PBMCs were viably cryopreserved for later experimentation. Biospecimens were maintained by the Melbourne Children's Bioresource Centre with facility staff carrying out sample processing, tracking, and long-term liquid nitrogen storage following international best practice.

**Isolation, activation, and expansion of naive CD4+ T cells**. Cell cultures were set up in a blinded fashion. Viable total CD4+ T cells were identified, Tregs (CD3 +CD4+CD25+CD127−) were excluded and naive CD4+ T cells (CD3+CD4 +CD25−CCR7+CD127−) were purified by flow cytometry. To do this, PBMCs were first thawed using warm RPMI media (Sigma-Aldrich) supplemented with 10% heat-inactivated fetal calf serum (FCS) (HyClone), centrifuged at $500 \times g$ and washed twice before viability assessment by Trypan blue on the TC20 automated cell counter (Bio-Rad). Mean PBMC viability was 87% across all subjects, 82% across allergic subjects, 86% across non-allergic controls, and 89% across resolved allergics. Cell pellets were resuspended at $1 \times 10^6$/ml in phosphate-buffered saline (PBS) and 0.5 µl of fixable viability stain 510 (BD Biosciences) added per ml of cell suspension. Cells were incubated at room temperature for 15 min protected from light, washed twice in FACS buffer (2% FCS, 2 mM EDTA in PBS), and resuspended in 50 µl FACS buffer for antibody staining. The antibody cocktail used to define naive T cells included CD3-APCH7, CD4-A700, CD25-PE, CD127-V450, CCR7-PECF954, and CD45RA BV711. All antibodies and isotype controls were purchased from BD Biosciences (Australia). Cell sorting was performed using an Influx Cell Sorter (BD Biosciences). Ex vivo viable naive CD4+ T cells were seeded in 96-well round bottom plates at a density of $8 \times 10^4$ cells per well in a total volume of 200 µl of RPMI media supplemented with 10% FCS, 100 U/ml penicillin–streptomycin, and 2mM L-glutamine + rhIL-2 (200 U/mL, Life Technologies). T cells were divided into half and either activated with 2 µl of Human T cell activator CD3/CD28 Dynabeads (Thermo Fisher Scientific) per well (1:1 ratio bead-to-cell) or left resting in media alone for 72 h at 37 °C and 5% $CO_2$. At culture end-point, cells were thoroughly resuspended and magnetic beads removed prior to obtaining cell and viability counts by Trypan blue exclusion on the TC20 automated cell counter. T cell proliferation was determined as the magnitude of the difference between stimulated and un-stimulated control wells at 72 h.

**Isolation and quantitation of nucleic acids**. At termination of cultures, activated and resting cells were recovered and centrifuged at $500 \times g$ for 5 min in RPMI media and cell pellets lysed in 350 µl of RLT buffer from the Qiagen AllPrep DNA/ RNA Extraction Micro Kit (Qiagen). Genomic DNA and total RNA were extracted according to the manufacturer's instructions using the QIACube automated liquid handling robot (Qiagen). DNA and RNA were quantified on the Qubit flourometer using the Qubit dsDNA Broad Range Kit or the Qubit hsRNA Kit (Thermo Fisher Scientific).

**Quantitation of secreted cytokines**. Cell culture supernatants were harvested and frozen at −80 °C for later quantification of TNF, IL-2, IFN-γ, IL-10, IL-6, IL-4, and IL-17A by cytometric bead array (Human Th1/Th2/Th17 Kit, BD Biosciences) according to the manufacturer's instructions. CBA data were acquired on an LSR II X-20 Fortessa (BD Biosciences) and analyzed using the LEGENDplex software (BioLegend). The limits of detection for all assays was 20 pg/ml. Sample values below the limit of detection (LOD) were set to LOD/2.

**Genome-wide profiling of DNA methylation**. Genomic DNA (200 ng) from patient naive T cell samples were randomized and sent to Service XS (Netherlands) for sodium bisulfite treatment and genome-wide methylation analysis using Illumina InfiniumMethylationEPIC BeadChips, which enables methylation measures at over 850,000 CpG sites. Raw.iDAT files were preprocessed using the Minfi package[44] from the bioconductor project (http://www.bioconductor.org) in the R statistical environment (http://cran.r-project.org/, version 3.3.0). Sample quality was assessed using control probes on the array with one sample removed due to poor assay performance. Between-array normalization was performed using the stratified quantile method to correct for Type 1 and Type 2 probe bias[45]. Probes exhibiting a $P$-detection call rate of >0.01 in 1 or more samples were removed (14,036 probes) prior to analysis. Probes containing SNPs at the single base extension site, or at the CpG assay site were removed, as were probes measuring

non-CpG loci (32,445 probes). Probes reported to have off-target effects in McCartney et al.[46] were also removed (39,690). After sample and probe filtering the final data set size was 210 samples and 780,665 probes. Methylation percentages were derived as β values with log 2 transformation to $M$ values for statistical analysis[47]. Batch correction was applied to $M$ values using the ComBAT function in the SVA package[48].

**RNA-sequencing**. Total RNA was randomized and sent to the Translational Genomics Unit—Sequencing Service and Development Platform at the Murdoch Children's Research Institute/Victorian Clinical Genetics Services for next-generation sequencing. Library preparation was performed using the Illumina TruSeq Stranded mRNA Kit with a starting input of 50 ng and libraries were sequenced on the Illumina HiSeq 4000 instrument generating an average of 20 million reads per sample. Fastq files were assessed for quality using the FastQC package[49] and no samples were excluded from analysis. The Salmon aligner (v0.8.2,[50]) was used to map 75-bp paired-end reads to the human transcriptome (Ensembl, GRCh37v75). Uniquely mapped reads were summarized across all genes with the tximport program[51]. Genes not highly expressed were filtered out (<10 counts per million), and multidimensional scaling analysis was used to assess sample quality during which two samples were removed leaving a final data set size of 18,864 genes and 134 samples. The count data were TMM normalized[52] and voom transformed prior to statistical analysis[53].

**Genotyping**. Existing SNP-array genotyping data were available for subjects in this study as part of a previous investigation into the genetic determinants of food allergy[54]. These individuals were genotyped on the Illumina Human Omni2.8v8 array. Individuals were excluding if sample call rates were <95%. SNPs were excluded if call rates were <95% or allele frequencies were <5% or failed the test for Hardy–Weinberg equilibrium ($P < 0.001$) or had significant ($P < 0.001$) differential missingness between cases and controls. In total 42/65 individuals in this study had high-quality genotype data available. SNPs were selected for association testing by virtue of being annotated to a heuristic ± 10 kb window of the food allergy genes of interest, or within the de novo DNA methyltransferase enzymes, proving coverage across the entire gene sequence[25]. The number of SNPs covering each gene can be found in Table S6. For selecting SNPs previously associated with food allergy in SNPedia, we used proxies annotated in the SNAP database[55] with an $r^2$ cut-off of 0.5 that were genotyped on the Omni array and passed QC. We also queried the AIRES GCTA database[27] across all time points using the default trans distance of 1 Mb for potential mQTLs and selected proxies using SNAP. Case–control association testing was carried out in PLINK[56] under an additive model, adjusting for ancestry, with Bonferroni correction applied to limit Type 1 error. Ancestry variables (Caucasian, non-Caucasian) were derived from parent report and validated by genetic inference using identity-by-state analysis and visual inspection of multidimensional scaling analysis of all individuals and reference individuals from the 1000 Genomes project[54].

**Statistical analysis**. The statistical analysis for this study was pre-registered with the Center for Open Science (https://osf.io/pys9e/register/ 565fb3678c5e4a66b5582f67) and the project is deposited in the Open Science Framework repository (https://doi.org/10.17605/OSF.IO/HBWRE). For hypothesis testing (differential analysis) of methylation and gene expression data, a factorial regression model was fitted to the data with adjustment variables for batch (flow cell or chip position), sex, ancestry, eczema status, and the first seven PCs. As this was a repeated-measures design, each subject was treated as a random effect. Cross-sectional comparisons were made at baseline and follow-up, and for the longitudinal analysis we constructed a model that included main effects for treatment (activated or quiescent), time, allergy status, and fitted an interaction term for time × allergy status. All between-group comparisons were made using moderated $t$ tests from the R Bioconductor limma package[57]. Evidence for differential methylation/expression was declared when false discovery corrected $P$ values were 0.05 or lower. For comparisons of activated vs. quiescent T cells, we applied an additional threshold filter to remove genes with a log fold change <1.5. We used QQ plots and calculated the genomic inflation (lambda) factor to determine how well the model fit the data and to check for any potential sources of bias, and these data are provided in Supplementary Figure 7. For integrated analysis of differentially methylated loci and gene expression, genomic coordinates of differentially methylated probes mapping to human genome build 19 (hg19) were extracted from the manufacturer's annotation file using the "getAnnotation" function in the Minfi R package. For gene expression, ENSMBL transcript ID's were mapped to hg19 using the org.Hs.eg.db in Bioconductor. The resulting genomic coordinates and differential analysis statistics were converted to GRanges objects and the data sets were merged by overlapping coordinates using the "mergeByOverlaps" function in the IRanges r package[58]. Gene set testing was carried out using the Camera competitive test accounting for inter-gene correlation using 0.01 as input parameter[59]. Publicly available ChIP-seq and DNAse1 accessibility data were downloaded from the ENCODE[60] and ROADMAP[61] consortiums. Enrichment analysis of these data were carried out using the Genomic Hyperbrowser GSUITE of tools. A suite of tracks representing different genomic features for a specific cell type were selected from the ENCODE repository. To determine which tracks in the suite

exhibit the strongest similarity by co-occurrence to experimentally determined regions of differential methylation, the Forbes coefficient was used to obtain rankings of tracks, and Monte Carlo simulations were used to define a statistical assessment of the robustness of the rankings using randomization of genomic regions covered by the EPIC array to derive a null model, and compute test statistics[62,63]. Methylation and expression QTL analysis was performed using the Matrix eQTL R package[64] under a linear additive model and a homoscedastic error covariance matrix.

**Data availability**. The data sets generated and analyzed for the current study are deposited in the Gene Expression Omnibus repository with the primary accession code GSE114135.

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

## Acknowledgements

We wish to acknowledge the contribution of the HealthNuts investigators Shyamali Dharmage, MBBS, MSc, Melissa Wake, MD, FRACP, Lyle Gurrin, PhD, Anne-Louise Ponsonby, MBBS, FAFPHM, FRACP, PhD, Adrian Lowe, PhD, Terrance Dwyer, PhD, and Melanie Matherson, PhD. We wish to thank the families and study participants involved in the HealthNuts study. Leone Thiele, BA, MNSc, Helen Czech, RN, Jeeva Sanjeevan, MBBS, Marnie Robinson, MBBS, FRACP, Dean Tey, MBBS, FRACP, Hern-Tze Tina Tan, PhD, Deborah Anderson, RN, and Giovanni Zurzolo, BSc, contributed to collection of data through recruitment of infants in immunization sessions and/or food challenges. Belinda Phipson and Jovana Maksimovic provided advice on bioinformatics.

This work was supported by the National Health and Medical Research Foundation (GNT1084017), the DHB Foundation (CT21242), and the Victorian Government's Operational Infrastructure program.

## Author contributions

R.S. is principal investigator on this study and provided significant intellectual input into the conception and design of the study, as well as obtaining funding. Subject recruitment, data collection, and food allergy phenotyping of the HealthNuts study was overseen by K. A., M.T., and P.V. and conducted by a team of investigators. They also provided significant clinical input into patient selection and interpretation. Laboratory experiments and data collection were overseen by D.M. with assistance from M.N, T.D., and A.B. Genetic analyses was carried out by J.C. and J.E. D.M. is primarily responsible for generation of DNA methylation and RNA-seq data sets, data cleaning, and analysis. D.M. wrote the manuscript and all co-authors reviewed the manuscript and provided input into the final draft.

## Additional information

**Competing interests:** The authors declare no competing interests.

