## [Peer Review File · Nature Communications]

Reviewers' comments:

Reviewer #1 (Remarks to the Author):

This study follows on from a 2012 publication in the journal *Allergy* by the first author describing that there are differences in the proliferative potential of naive T cells from allergic vs non allergic children when the T cells are stimulated by poly clonal activators such as Anti-CD3. In this submitted manuscript the authors use transcriptional profiling and epigenetic analysis to probe for the potential nature of the deficiency in proliferative potential in polyclonal activated CD4 T cells using comparative transcriptional profiling and epigenetic studies of both naive and polyclonal activated CD4 T cells from 34 milk allergic children vs naive and activated CD4 T cells from 21 non milk allergic children. The key finding in this submitted manuscript is in figure 3 and 4A, where the authors identify 24 genes that are differentially expressed and methylated between activated T cell of allergic and non allergic children.

When they compared the changes seen in T cells from children that resolved allergy and those that are still allergic, these 24 genes are now similar in expression and methylation patterns (figure 4b) between the two groups. The authors argue that the the propensity to develop an allergic phenotype in children must relate to the differences in gene expression of the metabolic related genes seen at the activated naive T cell level. However an alternative explanation is that the epigenetic changes in these 24 genes were simply not stable over time or reflect some bystander confounding mechanism related to the donors status and the lack of difference in these 24 genes when comparisons are made at the activated T cell level have little relevance to the development of allergen specific CD4 T cell phenotype. Further, although the transcriptional/epigenetic changes reported in the manuscript are appropriately performed there is insufficient data either proliferation or metabolic studies on the in vitro model the authors use to convince this reviewer that the naive T cells from allergic individuals and certainly the allergic T cells that are pathogenic for for allergy are in any way metabolically different from the non allergic T cells that become active. Also the lack of time course proliferation data using CSF is critical as it is impossible to determine just what is the status of the responding cell types from allergic or non allergic cultures.

Therefore, although the description of these genetic differences could be interesting, they are only a descriptive part of the story that requires more in depth validation of the in vitro activation model so that the gene changes/pathways described by the authors can be considered as predispose to either a Th2 or a non-Th2 phenotype. In short there is still a great deal of work to be done by the authors to support the title and the stated claims of the manuscript, namely that they have identified dysregulated metabolic pathways in the bulk naive CD4T cell of individuals who are predisposed to developing allergy.

Reviewer #2 (Remarks to the Author):

The manuscript entitled, "Epigenomic profiling of naïve CD4+ T-cells identifies dysregulated cell cycle and metabolic pathways in childhood food allergy", by Martino et al., seeks to determine whether there are CD4+ T-cell precursor cell intrinsic differences in response to antigen presentation among individuals with egg allergy compared to individuals without egg allergy. The authors report differences in cell proliferation and epigenomic patterns between quiescent and activated CD4+ T cells, differences between allergic and non-allergic individuals for the quiescent and activated cell states, as well as subsequent molecular profiles of children with food allergy at age 2 or 4. Epigenomic analysis of all quiescent and activated cells revealed a core set of 59 differentially remodeled genes associated with an activated state. Several known early T helper effector cell surface markers showed increased gene expression in activated cells relative to quiescent cells. The authors then compared cell proliferation, cell viability, and epigenomic patterns in activated cells and found differences between individuals with and without food allergy. More specifically, cell proliferation was shown to be reduced in the allergic group compared to the non-allergic group, IL-10 and IFN-gamma cytokines production was increased among allergics

upon activation, differential expression of cell cycle, signal transduction, and metabolic genes, and substantial loss of methylation at 89% of differentially methylated loci was observed in activated cells obtained from individuals with egg allergy compared to non-allergic individuals. A core set of 24 genes showed both significant DNA methylation and gene expression changes. The authors address whether the epigenetic changes they observed in 24 core genes associated with food allergy are controlled by SNPs located in the same genes and report no evidence of genotype controlling their egg allergy related epigenetic changes. Finally, among allergic cases and a subset of food-allergy candidate genes (identified above), the authors compared 2-4 year old epigenetic profiles of children that developed tolerance to egg compared to those that were still allergic to egg at age 2 or 4. They found the epigenomic profiles of children that developed tolerance were more similar to unaffected children at age 1 while 2-4 year olds with persistent egg allergy had profiles more similar to 1 year old allergic individuals.

This study addresses an important gap in our knowledge with respect to differences in CD4+ T cell activation states among individuals with egg allergy. It has several notable strengths including examination of a homogenous precursor cell with direct relevance to the disease phenotype and longitudinal measurements from the same individuals, with some developing egg tolerance by age 2-4. However, there are several issues that put the findings into doubt that should be addressed, they are:

Major concerns:

1) The authors report "Widespread loss of methylation was shown to be enriched at active enhancer regions", referenced on line 134. What was the background reference of null regions used to compute enrichment statistics? It appears to be all track elements in the genome but should be restricted to those measured on the EPIC array.

2) The authors report "Differential methylation was enriched within the gene body...", referenced on line 206 and Figure S4, as well as "methylated regions (allergy-dmrs) were enriched at lineage-defining...", referenced on line 210. What was the background reference of null regions used to compute enrichment statistics? For both it should be regions and chromatin regions measured on the EPIC array as opposed to the whole genome.

3) There are several technical issues with the 24-gene genetic analysis that put the results into question and need to be addressed:

(a) ancestry is not appropriately controlled via self-reported ancestry information; it could result in type I and type II errors. Previous studies have shown self-report does not completely address potential confounding by genetic ancestry in genetic studies (e.g. PMID: 15941970). The authors state genome-wide genotyping measurements were obtained; these could be used to generate eigenvalues for use as covariates in the food allergy genetic analysis (see PMID: 16862161).

(b) the authors state that the criteria for including SNPs to test was being located within a 10kb window. Instead of using an arbitrary physical distance selection criteria for the 24 genes under investigation, they should consider criteria based on linkage disequilibrium blocks and select tag SNPs for testing. I did not find a description of how the SNPs were selected other than physical distance nor did I find a description of how many genes had SNP representation or how well the tested SNPs covered genetic variation in these candidate regions.

(c) the authors perform genetic association testing to identify SNPs associated with egg allergy in their sample and only include SNPs in close local proximity to the food allergy associated differentially methylated and expressed genes. While I understand the need to limit the number of SNPs tested due to a very small sample size, they should also consider testing previously reported food allergy associated SNPs (a relatively small list) directly or via suitable tag SNPs that lie outside of the 10kb window to determine whether they control DNA methylation at these newly discovered 24 core food-allergy associated genes.

(d) although this study is limited to CD4+ T cells, it may be useful to use publically available SNP meQTL-DNA methylation/expression maps from other large blood datasets given the very small sample available in this study. These may provide insights into which SNPs (outside of the 10kb window) could control potentially control methylation/expression levels at the 24 genes under investigation and thus provide additional SNPs for food allergy association testing.

(e) given the small sample size and limited assessment of genetic variation the statement on line 269-270 reading, "loss of methylation at RPTOR associated with food allergy was independent of genetic variation" seems a bit of a stretch to conclude since it was not comprehensively assessed.

The relationship to underlying genetic variation, and these points above, are particularly important given the large difference in family history of food allergy between cases and controls that is shown in Table S1.

4) It is difficult to interpret the findings as presented in Figure 2A and 3C. A volcano plot would be more useful for assessing meaningful changes; the level of significance versus effect size is not presented in the circle plots and thus can't be assessed.

5) QQ plots are not presented for the food allergy vs non-food allergy findings in either the quiescent or activated cell states. There could be inflation or deflation that impacts the interpretation of the differential methylation and expression findings.

6) Are the changes in gene expression profiles related to activated and quiescent cell states driven by or related to food allergy status? If one group of cells is driving this signature it may not be representative of T cell activation in among individuals, generally. Thus, it is important to know to interpret the results. For example, if you were to use unsupervised methods to cluster samples based on differentially expressed and methylated genes/loci identified as associated with an activated cell state versus quiescent cell state would they separate FA cases from controls?

7) The results shown in Figure 4 do not make sense to me, perhaps the titles are incorrectly labelled? The direction of effect seems to be opposite of what I expect. If the values really represent persistent v resolved shouldn't they be in the same direction as the allergic v non-atopic control plot in panel A, i.e. the persistent look more like allergic at baseline and resolved look more like controls at baseline?

8) The mean, median, and IQR differences in the graphs (Figures 2, 3, 4, S3) can't be interpreted because they are presented as filled in rectangles without clearly showing the median or mean values. A standard box and whisker plot and/or strip chart would be more useful, especially given the relatively small sample size. It would also help to have corresponding text in the results reporting the median number of quiescent cells present in FA and NA as well as for the activated cell states of each.

9) Line 104 in the first results section states "72 hours is optimal time...T-cells can rest in culture before significant culture -induced changes in DNA methylation are detected in genome-wide scans". How was genome-wide significance assessed? It would also be helpful to know what proportion of measured loci showed up to a 20% change in DNA methylation. Also, given the very small sample number, it would be helpful in Figure S1 to see 2 different colored points in the plot (or 1 plot per individual), one per individual.

10) The mean PBMC viability when thawed was reported to be 87% across all subjects in the Methods section (lines 401-402). Was there a difference in viability at the time of thawing between cells obtained from allergic individuals compared to controls? I assume the viability assays in the main figures were from the 72 hour time point which is different than this one.

11) Were the cell proliferation and viability assays performed in a blinded fashion, i.e. the experimenter could not tell from labels and did not know which cells were from allergic individuals versus controls? Were the cells balanced across the plate(s) with respect to the location of samples from allergics versus non-atopic controls?

12) Were samples randomized and/or balanced with respect to activated/quiescent and food allergy case/control status for DNA methylation and transcription measurements?

13) In the methods section, the authors state that Combat was used to adjust for batch effects (lines 449-450) and later state that the regression model was fitted to the data with adjustment variables for batch (line 479). Adjusting for batch 2 times could lead to genomic deflation. In addition, the authors state seven principal components were included as covariates. Were the principal components related to any known sources of biological variation? How many in total were identified and why were 7 chosen for inclusion?

14) A detailed description in the methods of how genes were selected for assessing the relationship between expression and methylation in Figures 2D and 3, is needed. It is impossible to interpret what the expected relationship between gene expression and DNA methylation without knowing how they were mapped and annotated to each another.

15) Figure 3a and the corresponding result text reports attenuated cell proliferation, based on cell counts, in cells obtained from allergic individuals compared to controls. It appears that the food allergy patients had fewer quiescent cells than the non-allergic controls to begin with. Is it not surprising then that they also have fewer cells after activation? I think perhaps that is what the fold change plot is trying to address but it is impossible to interpret without actual median/mean values for each group plotted or listed. How was fold change computed?

More minor concerns:

16) There was a lack of detailed figure and table legends and/or superscripts in many places which made it difficult to interpret the values and results. These include, but are not limited to:

a) Table S1 has a row labeled "Median age of egg introduction (m)". What is m? months? 2 months seems a bit early for solid food introduction based on current US recommendations?

b) Table S4 heading says "5-methyl cytosine" but the sodium bisulfite based detection method used can't distinguish between 5-methyl and 5-hydroxymethyl cytosine.

c) Tables S3, S4, S5 do not have a legend or superscripts describing what the reported values are. For example, in Table S4, there is a column labeled delta beta. I assume this is reporting the difference in DNA methylation between allergics and non-allergics but have no idea what direction the effect is, i.e. is it allergics-non-allergics or vice versa?

d) It would be more useful to have some of these tables in a .csv or .txt format so they can be easily sorted. Table S4 seems to be sorted by Gene Symbol which is not particularly useful when trying to find the most significant or greatest magnitude of change

e) Figure S4 is missing a legend

f) Figure S7. Unclear from the legend which EWAS comparison this is from

17) Line 138, I think the reference to statistics in Table S3 should actually reference Table S2.

18) I think line 225 seeks to reference Figure 3 and not Figure 2e, typo?

19) line 265 states "that methylation patterns at these 6 loci were methylation quantitative trait

loci (meQTLs)". This reads that the methylation loci are called meQTLs but the term me- or mQTL typically is what the SNP is called not the methylation target of the SNP.

20) It appears that in Table S1, the values reported for Average sIgE egg white Followup (kU/mL) in persistent and transient egg allergy cases may be flipped.

21) Table 1 provides results for gene set analysis of 72 hour quiescent versus activated T cells but a list of the differentially expressed and methylated genes/loci for the analytic comparison of all activated and quiescent cells, or even better would be all summary statistic results, should be provided as supplementary data.

Reviewer #3 (Remarks to the Author):

Decision on this manuscript: Rejection

The investigators analyze CD4+ T cell-responsiveness in a longitudinal fashion in food allergic infants. They perform a multilayer analysis using transcriptomics, epigenetic analysis and cytokine production levels (protein level). The major finding is T-cell hypo-responsiveness in food allergic individuals which can be related to cell cycles and metabolic pathways. With the development of tolerance in early childhood this hypo-responsiveness status is lost.

This is a retrospective study in which food allergic children (at an age of one year) are re-evaluated at an age of two to four years.

Although the study is very timely in terms of the integration of epigenetic and expression data in the field of allergy, there are a number of open and unclear issues which need to be resolved:

1. Since this is a retrospective analysis, it needs to be ensured that the data presented in this manuscript, i. e. DNA methylation, gene expression, and cytokine production have been generated for the purpose of this work. It is important that the major data presented in this manuscript are new. Furthermore, it is important to ensure the quality of the samples stored for apparently several years before analysis and further functional assays have been performed. Quality assurance between visit 1 (clinical assessment of food allergy) and visit 2 (re-evaluation) must be demonstrated.
2. It remains unclear why only mono-sensitized/allergic subjects are being involved. Are there major differences between mono- and poly-allergic individuals with regard to T cell-responsiveness?
3. How exactly is T-cell hypo-responsiveness defined? I am assuming a broad scatter of T-cell activation patterns in such a study group. Did the investigators use certain cut-off levels to discriminate hypo-responsiveness from "normal" responsiveness of CD4+ T cells? This needs to be precisely and clearly defined. Furthermore, how many food allergic individuals fulfil these criteria, assuming that not all food allergic infants respond the same way?
4. How was the T-cell activation actually conducted? I only read in material and methods (lines 412 and 413) the use of human T-cell activator dynabeads. What type of T-cell activation is this actually? And what is the read out for this activation procedure (T-cell proliferation as assessed with radioactivity or fluorochrome)? Quantitative assessment? Qualitative assessment?
5. Assuming that this method results in polyclonal, antigen-unspecific T-cell activation by directly cross-linking or activating the T-cell receptor, the question arises whether this state of "hypo-responsiveness" is only detectable in food-antigen-specific T cells or in virtually all T cells in this patient. Therefore it is important to discriminate between "bystander" hypo-responsiveness and hypo-responsiveness in the antigen-specific cells. This should have been easy to do since the study mainly focusses on egg-allergic individuals, so T-cell activation could have been performed also with respective egg antigen, such as ovalbumine. Maybe this has been done and the authors

should present data on this or at least thoroughly comment on this topic.

6. The observation of transcriptomic-epigenetic T-cell hypo-responsiveness and the relation to cell cycle and metabolic pathways is very interesting. But how does this relate to the development (or loss of (food) allergy)? Functional data are needed to provide this important missing link.

7. The authors have measured a broad panel of cytokines (lines 426 ff.), but they report only on a very limited number of cytokine results. All data need to be provided and discussed (at least in the supplement). Here again the spectrum and the scatter of responses should be shown as well. I am not assuming a homogenous response pattern in this patient population (and the follow-up as well).

8. Blood was collected one to two hours after the food challenge (line 391). Could it be that the food challenge triggers re-distribution of T cells to the site of reaction, so that the cells collected from the blood after the food challenge do not represent a normal distribution anymore? For example, a food challenge could result in recruitment of the food-specific T cells to the site of reaction and the cells present and left in the peripheral blood show this state of "hypo-responsiveness" because all the other cells are recruited to different anatomical sites. The authors need to demonstrate comparability of the results if blood was drawn before and after food challenges, at least in a small number of infants.

9. In the Introduction and the Discussion, please, refer to the recent state-of-the-art review on allergy epigenetics (<https://www.ncbi.nlm.nih.gov/pubmed/28322581>), especially while discussing CD4+ T-cell differentiation and its regulation by epigenetic mechanisms.

10. While discussing relative T-cell immaturities, please, refer to a very recently published study addressing this topic, also in the context of epigenetic mechanisms (<https://www.ncbi.nlm.nih.gov/pubmed/28159873>).

11. The word "remodeled" is used throughout the manuscript. Could you somehow replace it with something else? Remodeling has a certain biological meaning and although one can get what after reading the full manuscript it looks strange at first glance.

12. Lines 245-246. In the title of this subchapter, do you mean DNA methylation only or gene expression as well? It is unclear.

13. Tables 1, 2, and 4. Abbreviations below the table should be sorted either in the order of appearance or alphabetically.

14. Tables 1 and 2. "P.DE" does not appear in the table (probably you meant "P.DM").

Nevertheless, in the legend it is only said that it means the p-value. Thus, it is not necessary to introduce it at all, as it only creates confusion. Either replace it just with a "PValue" (or better "P-Value") or provide a better description if really relevant.

15. Tables 1-4. Abbreviations explained in one table should not be duplicated in the following tables. Better explain additional ones and then write "otherwise, please refer to Table XX" or something like that.

We wish to thank the reviewers for this thorough review of our manuscript. We note that reviewers 2 and 3 commented the study 'addresses important knowledge gaps', 'has several strengths' and 'is very timely' in terms of integrated multi-omic analysis. The revised manuscript addresses each reviewers concern point-by-point.

Reviewers' comments:

Reviewer #1 (Remarks to the Author):

This study follows on from a 2012 publication in the journal Allergy by the first author describing that there are differences in the proliferative potential of naive T cells from allergic vs non allergic children when the T cells are stimulated by poly clonal activators such as Anti-CD3. In this submitted manuscript the authors use transcriptional profiling and epigenetic analysis to probe for the potential nature of the deficiency in proliferative potential in polyclonal activated CD4 T cells using comparative transcriptional profiling and epigenetic studies of both naive and polyclonal activated CD4 T cells from 34 milk allergic children vs naive and activated CD4 T cells from 21 non milk allergic children. The key finding in this submitted manuscript is in figure 3 and 4A, where the authors identify 24 genes that are differentially expressed and methylated between activated T cell of allergic and non allergic children.

When they compared the changes seen in T cells from children that resolved allergy and those that are still allergic, these 24 genes are now similar in expression and methylation patterns (figure 4b) between the two groups. The authors argue that the the propensity to develop an allergic phenotype in children must relate to the differences in gene expression of the metabolic related genes seen at the activated naive T cell level. However an alternative explanation is that the epigenetic changes in these 24 genes were simply not stable over time or reflect some bystander confounding mechanism related to the donors status and the lack of difference in these 24 genes when comparisons are made at the activated T cell level have little relevance to the development of allergen specific CD4 T cell phenotype. Further, although the transcriptional/epigenetic changes reported in the manuscript are appropriately performed there is insufficient data either proliferation or metabolic studies

on the in vitro model the authors use to convince this reviewer that the naive T cells from allergic individuals and certainly the allergic T cells that are pathogenic for for allergy are in any way metabolically different from the non allergic T cells that become active. Also the lack of time course proliferation data using CSF is critical as it is impossible to determine just what is the status of the responding cell types from allergic or non allergic cultures.

Therefore, although the description of these genetic differences could be interesting, they are only a descriptive part of the story that requires more in depth validation of the in vitro activation model so that the gene changes/pathways described by the authors can be considered as predispose to either a Th2 or a non-Th2 phenotype. In short there is still a great deal of work to be done by the authors to support the title and the stated claims of the manuscript, namely that they have identified dysregulated metabolic pathways in the bulk naive CD4T cell of individuals who are predisposed to developing allergy.

Response: This study has characterized the key pathways of gene dysregulation in activated T-cells that now need to be explored in detail to determine the immunological consequences

for disease pathogenesis. It is one of the few studies to compare T-cell activation from a larger group of infants with challenge-proven, clinically relevant food allergy. Our study provides novel high quality important data regarding gene methylation and transcription in children with food allergy. We agree with Reviewer 1 that further studies are now needed to confirm related metabolic differences in naïve T cells in children with food allergy. We have modified the paper to ensure there is an appropriate emphasis on the epigenetic and transcriptomics findings, which are very strong, and less emphasis on the metabolic pathways in naïve T cell activation/differentiation, which are intriguing but require further work. We have made the following related changes:

1) New title: “Epigenetic dysregulation of naïve CD4+ T-cell activation genes in childhood food allergy”.

Changed from “Epigenomic profiling of naïve CD4+ T-cells identifies dysregulated cell cycle and metabolic pathways in childhood food allergy.”

-we feel the new title accurately reflects the data removes the emphasis on specific pathways which the reviewer felt more supportive data were needed.

2) New abstract: We have changed the conclusions to “Our data indicate epigenetic dysregulation in the early stages of signal transduction through the T-cell receptor complex in children with food allergy”, changed from “Our results suggest intrinsic dysregulation in cell cycle and metabolic pathways”.

3) New introduction and discussion: We have removed the paragraphs speculating on the links between naïve T-cell activation metabolism and the development of antigen specific populations. We have been careful to acknowledge that unknown clinical significance of these pathways (line 77/78). The discussion has been extensively revised to focus on immune development, and puts the findings within the context of well-known maturation differences in T-cell competence among children with allergic disease. We explicitly acknowledge the lack of CFSE and time-course data is a limitation of the study (line 668).

4) New Conclusion: that we have identified key pathways modified likely by gene-environment interactions that extend to T-cell activation genes.

Reviewer #2 (Remarks to the Author):

The manuscript entitled, “Epigenomic profiling of naïve CD4+ T-cells identifies dysregulated cell cycle and metabolic pathways in childhood food allergy”, by Martino et al., seeks to determine whether there are CD4+ T-cell precursor cell intrinsic differences in response to antigen presentation among individuals with egg allergy compared to individuals without egg allergy. The authors report differences in cell proliferation and epigenomic patterns between quiescent and activated CD4+ T cells, differences between allergic and non-allergic individuals for the quiescent and activated cell states, as well as subsequent molecular profiles of children with food allergy at age 2 or 4. Epigenomic analysis of all quiescent and activated cells revealed a core set of 59 differentially remodeled genes associated with an activated state. Several known early T helper effector cell surface markers showed increased gene expression in activated cells relative to quiescent cells. The authors then compared cell proliferation, cell viability, and epigenomic patterns in activated cells and found differences between individuals with and without food allergy. More specifically,

cell proliferation was shown to be reduced in the allergic group compared to the non-allergic group, IL-10 and IFN-gamma cytokines production was increased among allergics upon activation, differential expression of cell cycle, signal transduction, and metabolic genes, and substantial loss of methylation at 89% of differentially methylated loci was observed in activated cells obtained from individuals with egg allergy compared to non-allergic individuals. A core set of 24 genes showed both significant DNA methylation and gene expression changes. The authors address whether the epigenetic changes they observed in 24 core genes associated with food allergy are controlled by SNPs located in the same genes and report no evidence of genotype controlling their egg allergy related epigenetic changes.

Finally, among allergic cases and a subset of food-allergy candidate genes (identified above), the authors compared 2-4 year old epigenetic profiles of children that developed tolerance to egg compared to those that were still allergic to egg at age 2 or 4. They found the epigenomic profiles of children that developed tolerance were more similar to unaffected children at age 1 while 2-4 year olds with persistent egg allergy had profiles more similar to 1 year old allergic individuals.

This study addresses an important gap in our knowledge with respect to differences in CD4+ T cell activation states among individuals with egg allergy. It has several notable strengths including examination of a homogenous precursor cell with direct relevance to the disease phenotype and longitudinal measurements from the same individuals, with some developing egg tolerance by age 2-4. However, there are several issues that put the findings into doubt that should be addressed, they are:

Major concerns:

1) The authors report “Widespread loss of methylation was shown to be enriched at active enhancer regions”, referenced on line 134. What was the background reference of null regions used to compute enrichment statistics? It appears to be all track elements in the genome but should be restricted to those measured on the EPIC array.

Response: These enrichment analyses were carried out using the Gsuite hyperbrowser tool. At the time of analysis support for a restricted null model was not available, but through contacting the website developers, this feature has now been added. We have repeated our analysis using the EPIC array track elements as the null model as suggested. This analysis has not altered our main conclusions. The p-values and enrichment scores are different, and these have been amended in the revision, but the new analysis is supports the same conclusion as the previous manuscript i.e allergy dmrs were enriched in lineage defining chromatin regions in differentiated T cells.

The methods section outlining the statistical analysis (line 833) now state: “A suite of tracks representing different genomic features for a specific cell type were selected from the ENCODE repository. To determine which tracks in the suite exhibit the strongest similarity by co-occurrence to experimentally determined regions of differential methylation, the Forbes coefficient was used to obtain rankings of tracks, and Monte-Carlo simulations were used to define a statistical assessment of the robustness of the rankings using randomization of genomic regions covered by the EPIC array to derive a null model, and compute test statistics”

2) The authors report “Differential methylation was enriched within the gene body...”, referenced on line 206 and Figure S4, as well as “methylated regions (allergy-dmrs) were enriched at lineage-defining...”, referenced on line 210. What was the background reference of null regions used to compute enrichment statistics? For both it should be regions and chromatin regions measured on the EPIC array as opposed to the whole genome.

Response: This appears to the same issue as raised above. As noted, we have repeated these analysis with the EPIC array track elements and are main conclusion remains unaltered.

3) There are several technical issues with the 24-gene genetic analysis that put the results into question and need to be addressed:

(a) ancestry is not appropriately controlled via self-reported ancestry information; it could result in type I and type II errors. Previous studies have shown self-report does not completely address potential confounding by genetic ancestry in genetic studies (e.g. PMID: 15941970). The authors state genome-wide genotyping measurements were obtained; these could be used to generate eigenvalues for use as covariates in the food allergy genetic analysis (see PMID: 16862161).

Response: Ancestry for each individual was coded as 1=Caucasian 0=non-caucasian based on parent-report, and additionally validated by genetic determination of ancestry. For the latter, identity-by-state analysis was carried out and ancestry was determined by visual inspection of MDS plots for all individuals and samples of known ancestry from the 1000 Genomes project. Our coding of ancestry based on parent report was concordant with genetically inferred ancestry, as we have outlined previously in Martino et al, Clin. Exp All, 2017 where we performed a GWAS study.

In the revision, the methods section on genotyping now includes the following (line 800): “Ancestry variables (Caucasian, non-Caucasian) were derived from parent-report and validated by genetic inference using identity-by-state analysis and visual inspection of multi-dimensional scaling analysis of all individuals and reference individuals from the 1000 Genomes project.”

(b) the authors state that the criteria for including SNPs to test was being located within a 10kb window. Instead of using an arbitrary physical distance selection criteria for the 24 genes under investigation, they should consider criteria based on linkage disequilibrium blocks and select tag SNPs for testing. I did not find a description of how the SNPs were selected other than physical distance nor did I find a description of how many genes had SNP representation or how well the tested SNPs covered genetic variation in these candidate regions.

Response: There is no established rule for the selection of SNPs, and a strategy based on tag-SNPs that exploits local LD is one we have used before in de novo genotyping studies (Ashley et al, 2017, Allergy). This study made use of *existing* SNP array genotyping data, and the content on these SNP arrays are selected based on a tag-SNP strategy i.e the SNPs are selected by exploiting local LD. Our strategy was to test any SNP genotyped on the array within +/- 10kb of the genes of interest, and represents a simple and straightforward approach providing broad coverage across the entire gene sequence, originally outlined and justified in Peterson et al Plos One 2013. We acknowledge this approach does not address distant trans

associations. In the revised methods section, we have clarified this on line 791: “SNPs were selected for association testing by virtue of being annotated to a heuristic +/- 10 kb window of the food allergy genes of interest, or within the de novo DNA methyltransferase enzymes, proving coverage across the entire gene sequence²⁵. The number of SNPs covering each gene can be found in Table S6. For selecting SNPs previously associated with food allergy in SNPedia, we used proxies annotated in the SNAP database⁵⁵ with an r2 cut-off of 0.5 that were genotyped on the Omni array and passed QC. We also queried the AIRES GCTA database²⁷ across all time points using the default trans distance of 1Mb for potential mQTLs and selected proxies using SNAP.”

(c) the authors perform genetic association testing to identify SNPs associated with egg allergy in their sample and only include SNPs in close local proximity to the food allergy associated differentially methylated and expressed genes. While I understand the need to limit the number of SNPs tested due to a very small sample size, they should also consider testing previously reported food allergy associated SNPs (a relatively small list) directly or via suitable tag SNPs that lie outside of the 10kb window to determine whether they control DNA methylation at these newly discovered 24 core food-allergy associated genes.

Response: In the revision we have included this analysis by querying the GWAS and SNPedia catalogues for published food allergy variants (described for peanut allergy only) and selecting suitable proxies that were directly genotyped on our array using the SNAP database. These additional proxy SNPs were included in the association test although we did not find evidence for an association with egg allergy in this cohort. These analyses are described on line 383 and detailed in the methods (line 791) and the results are tabulated in Table S8.

(d) although this study is limited to CD4+ T cells, it may be useful to use publically available SNP meQTL-DNA methylation/expression maps from other large blood datasets given the very small sample available in this study. These may provide insights into which SNPs (outside of the 10kb window) could control potentially control methylation/expression levels at the 24 genes under investigation and thus provide additional SNPs for food allergy association testing.

Response: We input our genes of interested into the AIRES blood mQTL database which returned 10 snps potentially influencing methylation. We used SNAP to select suitable proxies that were directly genotyped and included these in the association analysis. We did not find evidence of an association at these new snps. Results are tabulated in Table S8.

(e) given the small sample size and limited assessment of genetic variation the statement on line 269-270 reading, “loss of methylation at RPTOR associated with food allergy was independent of genetic variation” seems a bit of a stretch to conclude since it was not comprehensively assessed. The relationship to underlying genetic variation, and these points above, are particularly important given the large difference in family history of food allergy between cases and controls that is shown in Table S1.

Response: We agree with this, and have modified the text accordingly:

Line 400 “we did not find evidence that loss of methylation at *RPTOR* associated with food allergy was substantially influenced by genetic variation within the SNPs tested in this cohort.”

We have added the following to the discussion (line 532): “. We note that 18% of our cases had a reported family history of food allergy, and thus we made use of available SNP-array genotyping data to perform an association analysis on a limited number of ‘modified’ food allergy genes but did not find strong evidence to suggest genetic variation in these genes was driving changes in DNA methylation... This should be interpreted cautiously in the context of limited study power, and further comprehensive studies in larger cohorts are required.”

4) It is difficult to interpret the findings as presented in Figure 2A and 3C. A volcano plot would be more useful for assessing meaningful changes; the level of significance versus effect size is not presented in the circle plots and thus can't be assessed.

Response: We chose circos plot visualizations for Figure 2A and 3C as they provide a convenient way to visualize both the DNAm and GE data sets on the same genome ideogram, allowing the reader to easily identify regions of similarity and difference, and resolve individual chromosomes. We experimented with including fold-change on the circos plot but this detracted from readability. A key feature in figure 3C is the large signal coming from chromosome 19 which would not be evident on a volcano plot. We agree volcano plots are useful for visualizing fold-change as well as significance, however we argue that ‘meaningful changes’ is a subjective judgement and implies that only large effect sizes are biologically relevant. Rather, biologically meaningful changes are those where *both* expression and methylation are perturbed, which can be visualized on the circos plot. Volcano plots were provided in Figure S7 in the original manuscript.

5) QQ plots are not presented for the food allergy vs non-food allergy findings in either the quiescent or activated cell states. There could be inflation or deflation that impacts the interpretation of the differential methylation and expression findings.

Response: QQplots were provided in the supplement (Fig S7) and as the methods state, we calculated the genomic inflation factor for each genome association test and found no evidence of genomic inflation/deflation.

6) Are the changes in gene expression profiles related to activated and quiescent cell states driven by or related to food allergy status? If one group of cells is driving this signature it may not be representative of T cell activation in among individuals, generally. Thus, it is important to know to interpret the results. For example, if you were to use unsupervised methods to cluster samples based on differentially expressed and methylated genes/loci identified as associated with an activated cell state versus quiescent cell state would they separate FA cases from controls?

Response: We thank the reviewer for this helpful suggestion. We performed the suggested analysis and found that unsupervised clustering of samples based on the T-cell activation loci does separate our case control groups along PC2, but it is mostly evident in the activated

cells. This supports our contention that gene dysregulation associated with food allergy is revealed by activation. Moreover, we also find that food allergy loci partially overlap T-cell activation genes, thus the reviewers' suggestions have helped refine the interpretation.

In the revision we have now included this evidence as a new figure 3. Line 249 references this figure in the main text: "Unsupervised principal component (PC) clustering analysis of the 4,154 differentially expressed genes, and the 558 differentially methylated loci revealed clustering according to both T-cell activation status (PC1) and food allergy status (PC2) (**figure 3A and figure 3B**). Clustering by food allergy status was substantially more evident among activated cells, although separation of allergy clusters was incomplete. This suggested that T-cell activation gene networks were at least partially related to food allergy status."

7) The results shown in Figure 4 do not make sense to me, perhaps the titles are incorrectly labelled? The direction of effect seems to be opposite of what I expect. If the values really represent persistent v resolved shouldn't they be in the same direction as the allergic v non-atopic control plot in panel A, i.e. the persistent look more like allergic at baseline and resolved look more like controls at baseline?

Response: We thank the reviewer for noting these issues with Figure 4. We have re-assessed our data and agree that our initial interpretation regarding the direction of effect was incorrect - we assumed the resolved allergics were getting better with time, when in fact the persistent allergics were getting worse.

In our longitudinal analysis of the 26 allergy loci, the trajectory of the change over time in DNA methylation levels from baseline to followup differs between allergics who resolve and those who don't. We observe that most of the 26 loci exhibit a statistically significant change in methylation status from BL to FU among activated cells of persistent allergics, but not resolved allergics. This suggests a cumulative increase in epigenetic disruption which is not seen in the resolved group. The cell count data reflects this as an even more attenuated capacity to divide among persistent allergics, which we originally mis-interpreted as the resolved allergics were improving. Our revised interpretation is that epigenetic changes associated with food allergy continue throughout childhood in the persistent allergics, but are stable in the resolved allergics, who achieve better lymphoproliferative responses. In the revision we have presented these data as interaction plots, which show the change in methylation over time between the two groups. We have amended the abstract (line 29):

"Infants who failed to resolve food allergy in later childhood exhibited cumulative increases in epigenetic disruption at T-cell activation genes and poorer lymphoproliferative responses compared to children who resolved food allergy", and discussion (line 536): "Moreover, the longitudinal analysis suggests the accumulation of epigenetic change throughout childhood uniquely among the group of children who failed to resolve food allergy, which would be inconsistent with genetic effects on methylation that are reported to be fairly stable over time"

8) The mean, median, and IQR differences in the graphs (Figures 2, 3, 4, S3) can't be interpreted because they are presented as filled in rectangles without clearly showing the median or mean values. A standard box and whisker plot and/or strip chart would be more

useful, especially given the relatively small sample size. It would also help to have corresponding text in the results reporting the median number of quiescent cells present in FA and NA as well as for the activated cell states of each.

Response: In the revised manuscript, we have presented the graphs of functional data as strip charts as suggested. In the text where between group comparisons are referenced we now quote the medians.

9) Line 104 in the first results section states “72 hours is optimal time...T-cells can rest in culture before significant culture –induced changes in DNA methylation are detected in genome-wide scans”. How was genome-wide significance assessed? It would also be helpful to know what proportion of measured loci showed up to a 20% change in DNA methylation. Also, given the very small sample number, it would be helpful in Figure S1 to see 2 different colored points in the plot (or 1 plot per individual), one per individual.

Response: We used an absolute change +/- 20% from the 0hr time point to broadly assess whether the methylome was changing simply by resting in cell culture i.e no formal statistical test was applied since there were 2 individuals assayed over multiple time points. We have removed the word ‘significant’ as it implies a statistical test and line 170 now reads: “In preliminary studies, we determined that 72 hours was an optimal time-point for methylation studies as this was the maximum time quiescent naïve T-cells can rest in culture before substantial culture-induced changes in DNA methylation are detected in the genome-wide scans (Fig.S1).”

The data are sufficiently clear by visual inspection of the provided MvA plots, which show that by 120hr there is substantial changes in methylation. This analysis in the supplement supports our choice of time point, but is peripheral to the main experiment.

10) The mean PBMC viability when thawed was reported to be 87% across all subjects in the Methods section (lines 401-402). Was there a difference in viability at the time of thawing between cells obtained from allergic individuals compared to controls? I assume the viability assays in the main figures were from the 72 hour time point which is different than this one.

Response: The reviewer is correct that the reported viabilities in the main figure are from the 72 hour time point. The post post-thaw PBMC viabilities were very similar between groups. In the revision, the methods section entitled **isolation, activation and expansion of naïve CD4+ T-cells** now states the following: “Mean PBMC viability was 87% across all subjects, 82% across allergic subjects, 86% across non-allergic controls and 89% across resolved allergics.”

11) Were the cell proliferation and viability assays performed in a blinded fashion, i.e. the experimenter could not tell from labels and did not know which cells were from allergic individuals versus controls? Were the cells balanced across the plate(s) with respect to the location of samples from allergics versus non-atopic controls?

Response: Experiments were performed in a blinded fashion with randomization across plates.

12) Were samples randomized and/or balanced with respect to activated/quiescent and food allergy case/control status for DNA methylation and transcription measurements?

Response: Yes, we took great care to randomize the genomics experiments to reduce avoid any confounding batch effects. In the revision we have now included the following in the methods section describing Genome-wide profiling of DNA methylation.

“Genomic DNA (200ng) from patient naïve T-cell samples were randomized and sent to Service XS (Netherlands)”

And in the methods section describing RNA sequencing

“Total RNA was randomized and sent to the Translational Genomics Unit – Sequencing Service and Development Platform at the Murdoch Children’s Research Institute/Victorian Clinical Genetics Services for Next Generation sequencing.”

13) In the methods section, the authors state that Combat was used to adjust for batch effects (lines 449-450) and later state that the regression model was fitted to the data with adjustment variables for batch (line 479). Adjusting for batch 2 times could lead to genomic deflation. In addition, the authors state seven principal components were included as covariates. Were the principal components related to any known sources of biological variation? How many in total were identified and why were 7 chosen for inclusion?

Response: The Illumina HumanMethylationEPIC array comes as a Sentrix slide or chip with 8 arrays positioned over the slide. As both chip and array position effects are common on this platform, we applied the combat function to remove technical noise attributable to Sentrix Chip ID as the batch variable. Prior to hypothesis testing we performed a principal components analysis of post-qc data and look for correlations between the PC’s and known clinical variables. In this analysis we identified that the top 7 PC correlated with cell activation status, Sentrix array position, gender, ancestry, whilst PC8 correlated with allergy status. Therefore in the regression model for case-control analysis includes the top 7 PC as surrogate variables, as well as the covariates listed in the methods section. With each model fit we examined genomic inflation, which is reported in the supplement, and we saw no evidence of deflation.

14) A detailed description in the methods of how genes were selected for assessing the relationship between expression and methylation in Figures 2D and 3, is needed. It is impossible to interpret what the expected relationship between gene expression and DNA methylation without knowing how they were mapped and annotated to each another.

Response: In the revision, we have included a description on the integrated analysis of DNA methylation and gene expression in the methods section entitled ‘**statistical analysis**’:

“For integrated analysis of differentially methylated loci and gene expression, genomic coordinates of differentially methylated probes mapping to human genome build 19 (hg19) were extracted from the manufacturers annotation file using the ‘getAnnotation’ function in the Minfi R package. For gene expression, ENSEMBL transcript ID’s were mapped to hg19 using the org.Hs.eg.db in Bioconductor. The resulting genomic coordinates and differential analysis statistics were converted to GRanges objects and the data sets were merged by overlapping coordinates using the ‘mergeByOverlaps’ function in the IRanges r package.

15) Figure 3a and the corresponding result text reports attenuated cell proliferation, based on cell counts, in cells obtained from allergic individuals compared to controls. It appears that the food allergy patients had fewer quiescent cells than the non-allergic controls to begin with. Is it not surprising then that they also have fewer cells after activation? I think perhaps that is what the fold change plot is trying to address but it is impossible to interpret without actual median/mean values for each group plotted or listed. How was fold change computed?

Response: Fold change after 72hrs activation, was calculated by subtracting the pre-activation cell counts from cell counts post activation. In the revision, the figure legends now clearly state

“Fold-change was calculated as post – pre-activation cell counts.” In the revision, the figures are presented with the medians reported in the text.

More minor concerns:

16) There was a lack of detailed figure and table legends and/or superscripts in many places which made it difficult to interpret the values and results. These include, but are not limited to:

a) Table S1 has a row labeled “Median age of egg introduction (m)”. What is m? months? 2 months seems a bit early for solid food introduction based on current US recommendations?

Response: Thank-you for pointing this out. Table 1 had an error and the revised manuscript now explicitly states ‘months’ and the values have been corrected. Average age of introduction was between 5-6 months for all groups.

b) Table S4 heading says “5-methyl cytosine” but the sodium bisulfite based detection method used can’t distinguish between 5-methyl and 5-hydroxymethyl cytosine.

Response: This has been changed and the title now reads
“Table S4 – Differentially methylated loci in food allergy”

c) Tables S3, S4, S5 do not have a legend or superscripts describing what the reported values are. For example, in Table S4, there is a column labeled delta beta. I assume this is reporting the difference in DNA methylation between allergics and non-allergics but have no idea what direction the effect is, i.e. is it allergics-non-allergics or vice versa?

Response: Supplementary tables now have superscripts describing abbreviations

d) It would be more useful to have some of these tables in a .csv or .txt format so they can be easily sorted. Table S4 seems to be sorted by Gene Symbol which is not particularly useful when trying to find the most significant or greatest magnitude of change

Response: We are happy to provide the tables in .csv format and have re-sorted Table S4 by adjusted P value.

e) Figure S4 is missing a legend

Response: This has been corrected as follows:

“Figure S4 – Overlap of differentially methylated CpG with genomic regions. Bar chart shows the percentage of food allergy associated differentially methylated regions stratified by genomic feature.”

f) Figure S7. Unclear from the legend which EWAS comparison this is from

Response: In the revision we have clarified this. The new legend reads:

“Figure S7 – Diagnostic plots of genome-wide data modelling at baseline in activated cells. (A) Manhattan of epigenome-wide association (DNA methylation data). (B) Volcano plot of effect sizes. (C) qqplot of pvalue distribution with genomic inflation factor. (D) Volcanoplot and qqplot of moderated t statistic for RNAseq analysis.”

17) Line 138, I think the reference to statistics in Table S3 should actually reference Table S2.

Response: This has been corrected

18) I think line 225 seeks to reference Figure 3 and not Figure 2e, typo?

Response: this has been corrected

19) line 265 states “that methylation patterns at these 6 loci were methylation quantitative trait loci (meQTLs)”. This reads that the methylation loci are called meQTLs but the term me- or mQTL typically is what the SNP is called not the methylation target of the SNP.

Response: The revised version now states, “indicating that methylation patterns at these 6 loci were under the influence of genetic variation.”

20) It appears that in Table S1, the values reported for Average sIgE egg white Followup (kU/mL) in persistent and transient egg allergy cases may be flipped.

Response: Thankyou for point this out. This has been corrected in the revision.

21) Table 1 provides results for gene set analysis of 72 hour quiescent versus activated T cells but a list of the differentially expressed and methylated genes/loci for the analytic comparison of all activated and quiescent cells, or even better would be all summary statistic results, should be provided as supplementary data.

Response: We omitted these gene lists as they are very large, (4000 + genes), but have now included the summary statistics and gene tables for RNAseq (new Table S2) and DNAm (new Table S3) datasets.

Reviewer #3 (Remarks to the Author):

The investigators analyze CD4+ T cell-responsiveness in a longitudinal fashion in food allergic infants. They perform a multilayer analysis using transcriptomics, epigenetic analysis and cytokine production levels (protein level). The major finding is T-cell hypo-responsiveness in food allergic individuals which can be related to cell cycles and metabolic pathways. With the development of tolerance in early childhood this hypo-responsiveness status is lost.

This is a retrospective study in which food allergic children (at an age of one year) are re-evaluated at an age of two to four years.

Although the study is very timely in terms of the integration of epigenetic and expression data in the field of allergy, there are a number of open and unclear issues which need to be resolved:

1. Since this is a retrospective analysis, it needs to be ensured that the data presented in this manuscript, i. e. DNA methylation, gene expression, and cytokine production have been generated for the purpose of this work. It is important that the major data presented in this manuscript are new. Furthermore, it is important to ensure the quality of the samples stored for apparently several years before analysis and further functional assays have been performed. Quality assurance between visit 1 (clinical assessment of food allergy) and visit 2 (re-evaluation) must be demonstrated.

Response: Data collected for this study was entirely new and specific for this investigation. The study design, data collection and analysis plan were pre-registered on the Open Science Framework prior to generation of the genomic and functional data. The time-stamped pre-registration (2016-10-27 23:33 UTC) and project can be found here, and this link has been added to the methods:

<https://osf.io/pys9e/register/565fb3678c5e4a66b5582f67>

We also have kept an evolving regular update of the project progress timeline on ResearchGate:

<https://www.researchgate.net/project/Methylation-sensitive-genes-in-food-allergy>

To ensure quality, our biospecimens were processed and maintained by the Melbourne Children's Bioresource Centre, with facility staff carrying out biospecimen processing, tracking and long-term liquid nitrogen storage following international best practice. As this was a longitudinal study we were able to compare post thaw viabilities, T-lymphoproliferative rates and cytokine responses for individuals with repeated measures and find the measures to be highly concordant between age 1 and age 2/4 samples. These additional data are provided with this rebuttal (see figure 1 in file named 'reviewers_supplement'), but we have opted not to include these data in the manuscript since it detracts from the flow of the paper.

In the revision we have included the following statement in the methods (line 705):

"The biospecimen collection was maintained by the Melbourne Children's Bioresource Centre, with facility staff carrying out biospecimen processing, tracking and long-term liquid nitrogen storage following international best practice." We have also included the link to our data analysis pre-registration.

2. It remains unclear why only mono-sensitized/allergic subjects are being involved. Are there major differences between mono- and poly-allergic individuals with regard to T cell-responsiveness?

Response: We chose to study mono-sensitized/allergic subjects for scientific clarity. As we examined T-cell responses during disease and resolution, data interpretation becomes difficult in poly-sensitized individuals who may resolve one food allergy but remained allergic to another etc. Clinically speaking, poly-sensitized individuals often represent the more severe end of the disease spectrum, but we have no specific data on mono- v poly-allergic T-cell responses.

3. How exactly is T-cell hypo-responsiveness defined? I am assuming a broad scatter of T-cell activation patterns in such a study group. Did the investigators use certain cut-off levels to discriminate hypo-responsiveness from "normal" responsiveness of CD4+ T cells? This needs to be precisely and clearly defined. Furthermore, how many food allergic individuals fulfil these criteria, assuming that not all food allergic infants respond the same way?

Response: T-cell hypo-responsiveness is a relative term and not defined in any formal way i.e we did not formally define 'responders' and 'non-responders'. We have used this term to refer to the diminished degree of responsiveness to physiological stimulation of naïve T cells seen in the allergic children. In the revision we show the individual data points for each functional measure, so the variability can be seen.

4. How was the T-cell activation actually conducted? I only read in material and methods (lines 412 and 413) the use of human T-cell activator dynabeads. What type of T-cell activation is this actually? And what is the read out for this activation procedure (T-cell proliferation as assessed with radioactivity or fluorochrome)? Quantitative assessment? Qualitative assessment?

Response: This was a classical polyclonal T-cell stimulation with co-stimulation (CD3/CD28) using bead-bound antibodies. As we purified naïve T-cells prior to seeding and activation, we determined proliferation rate as the difference in naïve T cell count between unstimulated and stimulated wells at 72hours, which was determined using an automated cell counter with staining for viability.

In the revision, we have included a more detailed description of the T-cell activation experiment in the introduction (line 93):

“We used genome-wide DNA methylation and transcriptional profiling to delineate molecular pathways of naïve T-cell responsiveness to activation under neutral (non-differentiating) conditions using bead-bound anti-CD3/anti-CD28 to polyclonally stimulate the canonical T-cell receptor signaling pathway.”

In addition, the methods now read (line 727):

“T-cells were divided in half and either activated with 2 μ of Human T-cell activator CD3/CD28 Dynabeads (Thermo Fisher Scientific) per well (1:1 ratio bead-to-cell) or left resting in media alone for 72 hours at 37°C and 5% CO₂. At culture end-point, cells were

thoroughly resuspended and magnetic beads removed prior to obtaining cell and viability counts by Trypan blue exclusion on the TC20 automated cell counter. T-cell proliferation was determined as the magnitude of the difference between stimulated and un-stimulated control wells at 72 hours.”

5. Assuming that this method results in polyclonal, antigen-unspecific T-cell activation by directly cross-linking or activating the T-cell receptor, the question arises whether this state of "hypo-responsiveness" is only detectable in food-antigen-specific T cells or in virtually all T cells in this patient. Therefore it is important to discriminate between "bystander" hypo-responsiveness and hypo-responsiveness in the antigen-specific cells. This should have been easy to do since the study mainly focusses on egg-allergic individuals, so T-cell activation could have been performed also with respective egg antigen, such as ovalbumine. Maybe this has been done and the authors should present data on this or at least thoroughly comment on this topic.

Response: We have not performed antigen-specific stimulation of naïve T cells, rather our study reports on suboptimal T cell proliferation that is generalized and polyclonal in nature. This may also extend to antigen specific clones. We disagree with the reviewer that ovalbumin specific T-cell activation experiments are easy. Our experience is that such studies are particularly challenging given the frequency of antigen specific naïve cells is very low in young infant blood, of which we collect 7mLs from each patient, yielding less than one hundred cells for this type of experiment. In the revision, in the discussion on study caveats we have included the following (line 6871):

“Our study is not able to address the question of whether T-cell activation is suboptimal in antigen-specific naïve T-cell populations, although we speculate that this is the case. Future studies should address this using peptide epitopes loaded onto major histocompatibility complex class II tetramers to address the function of antigen specific clones.”

6. The observation of transcriptomic-epigenetic T-cell hypo-responsiveness and the relation to cell cycle and metabolic pathways is very interesting. But how does this relate to the development (or loss of (food) allergy)? Functional data are needed to provide this important missing link.

Response: We agree with the reviewer that further functional data are now needed to establish the links between the pathways we have reported as dysregulated, and the development of food allergy. We speculate that what we are describing reflects maturational differences in immune development in children with food allergy and this will require a new cohort and is now the subject of ongoing investigation. Please refer to our comments in response to reviewer 1, and note that we have extensively revised the discussion.

7. The authors have measured a broad panel of cytokines (lines 426 ff.), but they report only on a very limited number of cytokine results. All data need to be provided and discussed (at least in the supplement). Here again the spectrum and the scatter of responses should be shown as well. I am not assuming a homogenous response pattern in this patient population (and the follow-up as well).

Response: In the main text of the original manuscript, we reported on the cytokines that show significant between group differences for readability, since the majority of cytokines

measured were below limits of detection, or not different between groups. In addition, we presented all cytokines measured for each individual as a heatmap in figure S2. In the revised version of the manuscript, we have redrawn the cytokine graphs in the main figures as scatterplots so individual data points can be clearly seen, and we have changed figure S2 from a heatmap to individual cytokine plots in response to this criticism.

8. Blood was collected one to two hours after the food challenge (line 391). Could it be that the food challenge triggers re-distribution of T cells to the site of reaction, so that the cells collected from the blood after the food challenge do not represent a normal distribution anymore? For example, a food challenge could result in recruitment of the food-specific T cells to the site of reaction and the cells present and left in the peripheral blood show this state of "hypo-responsiveness" because all the other cells are recruited to different anatomical sites. The authors need to demonstrate comparability of the results if blood was drawn before and after food challenges, at least in a small number of infants.

Response: We have addressed this by retrieving data pertaining to day of blood collection for the individuals in this study. We found that nine individuals in the study had bloods drawn on a different day to the egg challenge. We are able to compare proliferation and cytokine production between individuals whose bloods were collected after egg challenge against those with bloods drawn on a day of no egg challenge. We find no difference in any of the functional parameters testing, and are thus not concerned that blood drawn after challenge may be affecting our results. See figure 2 in the reviewer's supplement provided with this rebuttal for the data.

9. In the Introduction and the Discussion, please, refer to the recent state-of -the-art review on allergy epigenetics (<https://www.ncbi.nlm.nih.gov/pubmed/28322581>), especially while discussing CD4+ T-cell differentiation and its regulation by epigenetic mechanisms.

Response: Thankyou for bringing our attention to this oversight. This is now cited on line 60.

10. While discussing relative T-cell immaturities, please, refer to a very recently published study addressing this topic, also in the context of epigenetic mechanisms (<https://www.ncbi.nlm.nih.gov/pubmed/28159873>).

Response: The link provided does work but we assume this is the recent paper from the same group. This is now cited on line 71

11. The word "remodeled" is used throughout the manuscript. Could you somehow replace it with something else? Remodeling has a certain biological meaning and although one can get what after reading the full manuscript it looks strange at first glance.

Response: We have used the term 'remodeled' in reference to differentially methylated and expressed genes. Not only does it save space, it conceptually illustrates the underlying process of chromatin re-organization. Re-modelling has been used previously to describe changes in DNA methylation in Madlung et al, 2002.

12. Lines 245-246. In the title of this subchapter, do you mean DNA methylation only or gene expression as well? It is unclear.

Response: We have clarified this in the revision. The title now reads: *Polymorphisms at specific loci do not influence methylation or expression patterns at differentially remodeled genes*

13. Tables 1, 2, and 4. Abbreviations below the table should be sorted either in the order of appearance or alphabetically.

Response: Table abbreviations are now in alphabetical order

14. Tables 1 and 2. "P.DE" does not appear in the table (probably you meant "P.DM"). Nevertheless, in the legend it is only said that it means the p-value. Thus, it is not necessary to introduce it at all, as it only creates confusion. Either replace it just with a "PValue" (or better "P-Value") or provide a better description if really relevant.

Response: Thankyou legend for Tables 1 and 2 now amended.

15. Tables 1-4. Abbreviations explained in one table should not be duplicated in the following tables. Better explain additional ones and then write "otherwise, please refer to Table XX" or something like that.

Response: Thankyou we have taken this onboard.

REVIEWERS' COMMENTS:

Reviewer #1 (Remarks to the Author):

The changes submitted by the authors have resolved the issues that were raised in the original review.

Reviewer #2 (Remarks to the Author):

The revised manuscript now entitled, "Epigenetic dysregulation of naïve CD4+ T-cell activation genes in childhood food allergy", by Martino et al., characterizes differences in cell proliferation and epigenomic patterns between quiescent and activated CD4+ T cells, as it relates to egg allergy, as well as longitudinal molecular profiles of children with egg allergy at age 2 and 4. Several statistical and methodological concerns that were initially raised have all been addressed by the authors and their results remain unchanged.

Reviewer #3 (Remarks to the Author):

The following key points still require the generation of more data:

1. The manuscript is still too descriptive. The authors need to provide a mechanistic and functional link between T-cell responses, epigenetic changes, and food allergy as a clinical outcome.
2. It is still very unlikely that this state of a "hyperresponsiveness" affects all T-cells and T-cell subsets in T-cell subpopulations to the same degree. However, it is very likely that the antigen/allergen-specific T-cells behave differently in this regard. Therefore, the investigators need to demonstrate the functional and also the epigenetic status on these antigen-specific/allergen-specific T-cells although I realize that this is technically ambitious. However, without such additional datasets it is hard to explain the overall results.

Reviewer #1 (Remarks to the Author):

The changes submitted by the authors have resolved the issues that were raised in the original review.

Response: NA

Reviewer #2 (Remarks to the Author):

The revised manuscript now entitled, “Epigenetic dysregulation of naïve CD4+ T-cell activation genes in childhood food allergy”, by Martino et al., characterizes differences in cell proliferation and epigenomic patterns between quiescent and activated CD4+ T cells, as it relates to egg allergy, as well as longitudinal molecular profiles of children with egg allergy at age 2 and 4. Several statistical and methodological concerns that were initially raised have all been addressed by the authors and their results remain unchanged.

Response: NA

Reviewer #3 (Remarks to the Author):

The following key points still require the generation of more data:

1. The manuscript is still too descriptive. The authors need to provide a mechanistic and functional link between T-cell responses, epigenetic changes, and food allergy as a clinical outcome.
2. It is still very unlikely that this state of a “hyperresponsiveness” affects all T-cells and T-cell subsets in T-cell subpopulations to the same degree. However, it is very likely that the antigen/allergen-specific T-cells behave differently in this regard. Therefore, the investigators need to demonstrate the functional and also the epigenetic status on these antigen-specific/allergen-specific T-cells although I realize that this is technically ambitious. However, without such additional datasets it is hard to explain the overall results.

Response:

1. We agree further mechanistic work is needed but would also argue that functional and mechanistic links to clinical food allergy are self-evident in this study. We have uncovered a network of epigenetically regulated genes that are dysregulated in clinically food allergic children in association with deficiencies in T-cell activation. This suboptimal response is independent of antigen specificity, precedes the clinical manifestation of disease, and is therefore likely on the causal pathway. We have

Point-by-point response to reviewers

appropriately emphasized in the discussion that ongoing investigation is needed, as pointed out by the reviewer.

2. We agree the epigenetic status of T-cell activation genes in antigen-specific clones is needed, and we have clearly addressed this as a limitation of our study (line 468: “Our study is not able to address the question of whether T-cell activation is suboptimal in antigen-specific naïve T-cell populations, although we speculate that this is the case. Future studies should seek to address the epigenetic status of allergen-specific clones using peptide epitopes loaded onto major histocompatibility complex class II tetramers.”). However, we would also argue that our new data, in combination with our previous studies in monocytes (*Zhang, Sci Trans Med, 2017; Neeland, J Allergy Clin Immunol, 2018*) highlight the multicellular immunological deficits in food allergic individuals, beyond antigen-specific T cells. In our estimation this evidence that many aspects of immune function are altered suggests additional mechanisms important in the pathobiology of food allergy.